# The lncRNA *Neat1* promotes activation of inflammasomes in macrophages

Pengfei Zhang[1], Limian Cao[1], Rongbin Zhou [1], Xiaolu Yang[2] & Mian Wu[1,3]

The inflammasome has an essential function in innate immune, responding to a wide variety of stimuli. Here we show that the lncRNA *Neat1* promotes the activation of several inflammasomes. *Neat1* associates with the NLRP3, NLRC4, and AIM2 inflammasomes in mouse macrophages to enhance their assembly and subsequent pro-caspase-1 processing. *Neat1* also stabilizes the mature caspase-1 to promote interleukin-1β production and pyroptosis. Upon stimulation with inflammasome-activating signals, *Neat1*, which normally resides in the paraspeckles, disassociates from these nuclear bodies and translocates to the cytoplasm to modulate inflammasome activation using above mechanism. *Neat1* is also up-regulated under hypoxic conditions in a HIF-2α-dependent manner, mediating the effect of hypoxia on inflammasomes. Moreover, in the mouse models of peritonitis and pneumonia, *Neat1* deficiency significantly reduces inflammatory responses. These results reveal a previously unrecognized role of lncRNAs in innate immunity, and suggest that *Neat1* is a common mediator for inflammasome stimuli.

[1] The Chinese Academy of Sciences (CAS) Key Laboratory of Innate Immunity and Chronic Disease, CAS Center for Excellence in Cell and Molecular Biology, School of Life Sciences, University of Science and Technology of China, 230026 Hefei, China. [2] Department of Cancer Biology and Abramson Family Cancer Research Institute, Perelman School of Medicine, University of Pennsylvania, Philadelphia, PA 19104, USA. [3] Translational Research Institute, Henan Provincial People's Hospital, Academy of Medical Science, Zhengzhou University, Zhengzhou, Henan Key Laboratory of Stem Cell Differentiation and Modification, School of Clinical Medicine, Henan University, 450003 Zhengzhou, China. These authors contributed equally: Pengfei Zhang, Limian Cao. Correspondence and requests for materials should be addressed to X.Y. (email: xyang@mail.med.upenn.edu) or to M.W. (email: wumian@ustc.edu.cn)

  **1**

Inflammasomes are a group of multicomponent signaling platforms in the cytoplasm that control inflammatory response and anti-pathogen defense against a wide range of infection and damage signals[1–4]. These signals, including pathogen-associated molecular patterns (PAMPs) and damage-associated molecular patterns (DAMPs)[2,5], directly or indirectly activate one or more innate pattern recognition receptors (PRRs), which include nucleotide-binding domain (NBD) and leucine-rich repeat (LRR)-containing receptors (NLRs, e.g., NLRP1, NLRP3, and NLRC4), cytosolic DNA sensors (AIM2), and Pyrin (also known as TRIM20)[6–8]. Upon activation, the sensor proteins bind to and induce the oligomerization of a common adaptor protein, apoptosis-associated speck-like protein containing CARD (ASC), leading to the formation of a single macro-molecular aggregate known as ASC speck[9–11]. Oligomerized ASC recruits pro-caspase-1[12], facilitating its auto-processing into the mature subunits[13]. Active caspase-1 mediates proteolytic maturation of pro-inflammatory cytokines interleukin 1β (IL-1β) and IL-18 and elicits pyroptosis, a form of programmed cell death that exhibits features of both apoptosis (e.g., DNA fragmentation) and necrosis (e.g., plasma membrane rupture)[2,4,14–16].

While adequate inflammasome activation is crucial for the elimination of pathogens and damaged cells[17,18], dysregulation of inflammasome contributes to autoimmune, cancer, neurodegenerative disorders, and other diseases[19]. Nevertheless, the regulation of inflammasomes is not well understood. Several inflammasomes respond to a limited set of signals. For example, the AIM2 and NLRC4 inflammasomes are assembled upon the sensing of double-stranded DNA (dsDNA) and specific bacterial proteins, respectively[20,21], while the inflammasome formed with NLRP1 or its murine homolog Nlrp1b is activated by anthrax lethal toxin (LeTx) and 2-deoxy-D-Glucose (2DG)[22,23]. In contrast, the NLRP3 inflammasome is activated by an extraordinarily diverse array of PAMPs including several viral, bacterial, fungal pathogens, and DAMPs, such as crystalline, particulate (e.g., uric acid crystals, asbestos, and alum), extracellular ATP, pore-forming toxins, as well as change in cellular environment, notably hypoxia[5,24,25]. A salient unresolved issue is how various inflammasomes collectively are able to respond to such a wide spectrum of stimuli.

The majority of transcripts transcribed from human or mouse genome are non-coding RNAs[26,27]. Many are long non-coding RNAs (lncRNAs), which are defined as transcripts longer than 200 nucleotides but lacking significant protein coding capacity[28]. Thousands of lncRNAs have been identified to date[26,29–31], yet only a small fraction of them are characterized. In the context of innate immunity, while a few lncRNAs have been implicated in regulation of inflammasome, including LincRNA-Gm4419, LincRNA-Cox2, and lincRNA-EPS[32–34], none have been reported to directly involve in inflammasome assembly.

In the current study, we examine lncRNAs that associate with NLRP3 inflammasome in mouse macrophages and identify Neat1 (nuclear enriched abundant transcript 1), a lncRNA transcribed from the multiple endocrine neoplasia locus (hence also known as Men). Neat1 and its human ortholog NEAT1 maintain the structural integrity of the paraspeckles[35], a specific type of nuclear bodies in the interchromatin space whose function remains poorly understood[36]. NEAT1 also regulates the expression of a group of chemokines and cytokines, including IL-6 and CXCL10, through the MAPK pathway[37]. Of note, the expression of Neat1 is stimulated by many stimuli that also activate inflammasome, including infection of various viruses and some intracellular damages (e.g., ROS) that stabilize hypoxia-inducible factors (HIFs) and the tumor suppressor p53[38–40].

Here we find that Neat1 promotes the activation of NLRP3, NLRC4, and AIM2 inflammasomes and enhances caspase-1 activation, cytokine production, and pyroptotic cell death. Mechanistically, Neat1 binds to pro-caspase-1 and facilitates the assembly of inflammasomes, and also stabilizes the mature caspase-1 and increases caspase-1 protease activity. In response to various inflammasome-activating signals, Neat1 is released from paraspeckles and translocated to the cytoplasm to participate in inflammasome activity. Our findings establish a direct role for lncRNAs in regulating inflammasomes and suggest that Neat1 may represent a downstream convergence point for inflammasome stimuli.

## Results

**Neat1 enhances the activation of the NLRP3 inflammasome.** To investigate whether lncRNAs may regulate inflammasomes, we set out to identify lncRNAs that are associated with the NLRP3 inflammasome in murine immortalized bone marrow-derived macrophages (iBMDMs). We primed iBMDMs with lipopoly-saccharides (LPS) and subsequently treated them with the potassium ionophore nigericin to activate the NLRP3 inflammasome. After treating these cells with UV to stabilize protein-RNA interactions, we immunoprecipitated NLRP3 inflammasomes using anti-NLRP3 antibody and analyzed the associated RNAs by high-throughput sequencing (GSE118722). Several non-coding RNAs were found to be enriched in NLRP3 inflammasomes, including those encoded by Ankrd52, Hipk3, Pou2f2, Neat1, Gm26917, Gm42418, Snord34, Junos, Gm26767, and Gm23935 (Supplementary Fig. 1a). Among them were four lncRNAs: Gm26917, Gm42418, Gm26767, and Neat1 (Supplementary Fig. 1a).

To evaluate their effect on the NLRP3 inflammasome, we stably knocked down each of these four lncRNAs in iBMDMs using short hairpin RNA (shRNA). Knocking down Gm26917, Gm42418, or Gm26767 lncRNAs had no effect on the NLRP3 inflammasome (data not shown). Neat1 (Gene ID: 66961), is expressed in two isoforms, 3.2 kb Neat1_1 (NR_003513.3, also known as Men epsilon) and 20 kb Neat1_2 (NR_003513.3, also known as Men beta) (Supplementary Fig. 1b). We knocked down Neat1_1 and Neat1_2 simultaneously using two independent shRNAs (#1 and #2; Supplementary Fig. 1b, c). Interestingly, the knockdown of total Neat1 resulted in a strong reduction in oligomerization of ASC, indicative of impaired assembly of the inflammasome (Fig. 1a). This was accompanied by a noticeable reduction in the activation of caspase-1, processing of pro-interleukin 1β (pro-IL-1β) (Fig. 1a), and secretion of mature IL-1β (Fig. 1b, c). We also knocked down Neat1_2 alone with an isoform-specific shRNA (Supplementary Fig. 1b, d), and observed an almost indistinguishable set of results (Fig. 1d, e). Conversely, we forced the expression of Neat1. Neat1_2 is too long to be readily cloned and expressed. Therefore, we used the short variant Neat1_1 (Supplementary Fig. 1b). When Neat1_1 was over-expressed in iBMDMs (Supplementary Fig. 1h), LPS/nigericin-induced ASC aggregation, pro-caspase-1 cleavage, and IL-1β secretion were all increased (Figs. 1f, g).

Knocking down total Neat1 or Neat1_2 alone, or over-expressing Neat1_1, did not alter the expression of NLRP3, ASC, pro-caspase-1, or pro-IL-1β (Fig. 1a, d, f and Supplementary Fig. 1h), suggesting that Neat1 promotes the assembly of the NLRP3 inflammasome rather than increasing the expression of its components or downstream targets. These changes in Neat1 levels did not influence the secretion of TNF-α either (Supplementary Fig. 1e–g, i), indicating a specific effect of Neat1 on NLRP3 inflammasome-mediated cytokine production.

To verify the stimulatory effect of Neat1 on the NLRP3 inflammasome, we generated Neat1 knockout (Neat1$^{-/-}$) mice on the C57BL/6 background using CRISPR/Cas9-based genome

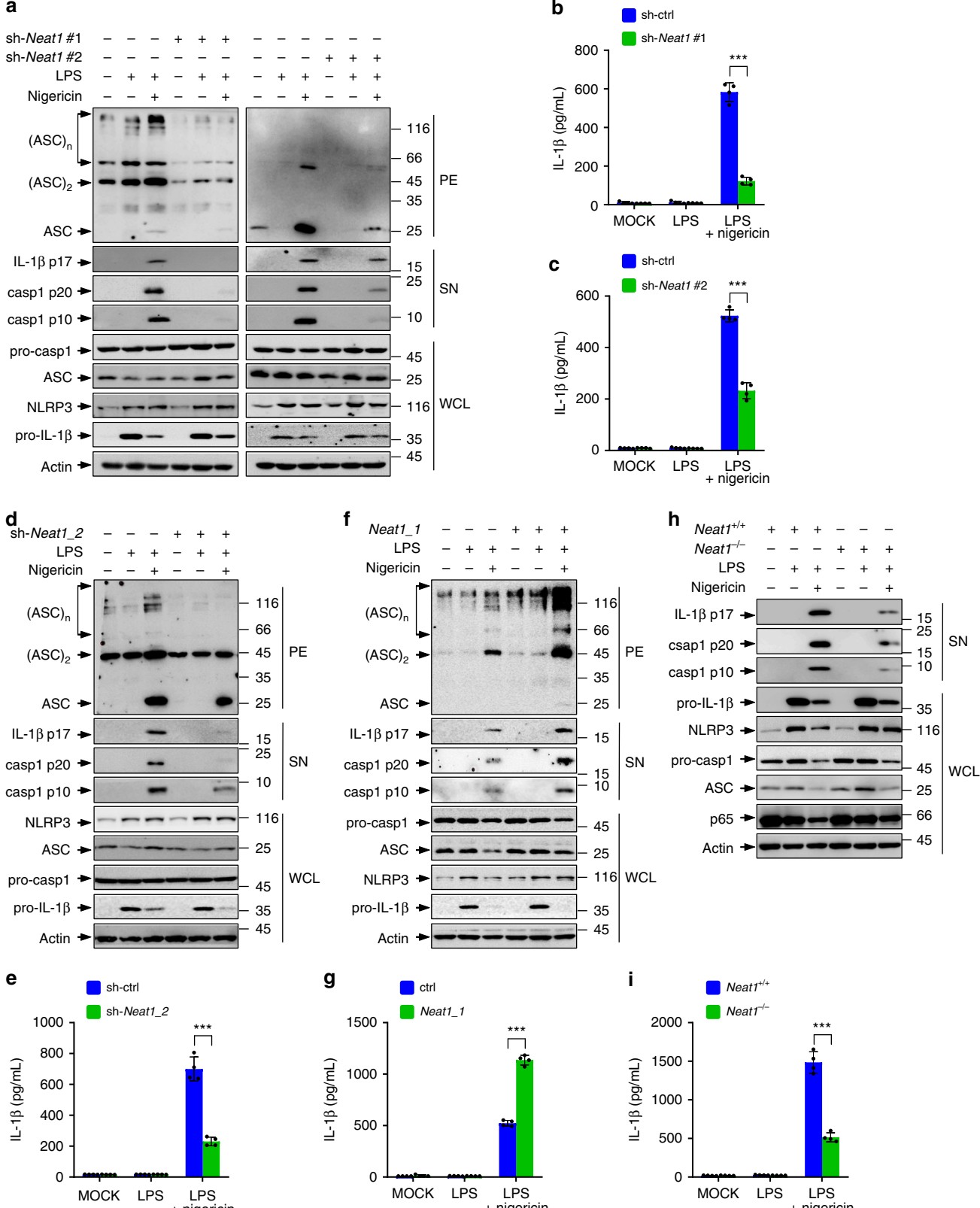

**Fig. 1** *Neat1* enhances the activation of the NLRP3 inflammasome. **a–c** Control iBMDMs and iBMDMs devoid of *Neat1* by shRNA #1 (**a**, left and **b**) or #2 (**a**, right and **c**) were untreated, primed with LPS, or treated with nigericin after LPS priming. Whole cell lysates (WCL) and the supernatant (SN) and pellet (PE) fractions were analyzed by western blotting for indicated protein levels (**a**). Culture supernatant was analyzed by ELISA for IL-1β secretion (**b**, **c**). **d–g** iBMDMs devoid of Neat1_2 (**d**, **e**) or overexpressing Neat1_1 (**f**, **g**) and the corresponding controls were treated with or without LPS or nigericin as indicated, and analyzed for protein levels (**d**, **f**) and IL-1β secretion (**e**, **g**). **h**, **i** BMDMs from *Neat1* wild-type (*Neat1*$^{+/+}$) or knockout (*Neat1*$^{-/-}$) mice were primed with LPS and then stimulated with nigericin as indicated. Protein levels (**h**) and IL-1β secretion (**i**) were analyzed. In **b**, **c**, **e**, **g**, and **i**, data shown are mean ± SD (*n* = 4). ***P < 0.001, two-tailed *t*-test. Source data are provided as a Source Data file

editing (Supplementary Fig. 1j, k). Upon co-stimulation with nigericin and LPS, BMDMs derived from $Neat1^{-/-}$ mice displayed much reduced caspase-1 activation and IL-1β secretion compared to BMDMs derived from wild-type ($Neat1^{+/+}$) mice (Fig. 1h, i), despite that both $Neat1^{+/+}$ and $Neat1^{-/-}$ BMDMs contained comparable mRNA and protein levels of NLRP3, ASC, pro-caspase-1, and pro-IL-1β (Fig. 1h and Supplementary Fig. 1k) and showed similar TNF-α secretion (Supplementary Fig. 1l). Moreover, $Neat1^{-/-}$ and $Neat1^{+/+}$ BMDMs also contained comparable levels of RelA/p65 (Fig. 1h and Supplementary Fig. 1k), an upstream activator of the NLRP3 inflammasome, further underscoring the specific effect of $Neat1$ on the NLRP3 inflammasome.

Consistent with its role in the assembly of the NLRP3 inflammasomes, overexpression of $Neat1\_1$ increased the interactions of NLRP3 with both pro-caspase-1 and ASC (Supplementary Fig. 1m). Therefore, $Neat1$ promotes the formation of the NLRP3 inflammasome, leading to enhanced caspase-1 activation and IL-1β maturation.

**$Neat1$ enhances activation of NLRC4 and AIM2 inflammasomes.** To evaluate the generality of the effect of $Neat1$, we tested NLRC4 and AIM2 inflammasomes, both of which mediate the activation of caspase-1. Murine macrophages were primed with LPS and then stimulated with flagellin or poly(dA:dT) to active the NLRC4 or AIM2 inflammasome, respectively. iBMDMs devoid of both $Neat1\_1$ and $Neat1\_2$ (Fig. 2a, b), or $Neat1\_2$ alone (Supplementary Fig. 2a, b), showed reduced activation of the NLRC4 and AIM2 inflammasomes, as indicate by a noticeable reduction in caspase-1 activation and IL-1β processing in response to flagellin and poly(dA:dT) treatment (Fig. 2a, b and Supplementary Fig. 2a, b). Conversely, iBMDMs with forced expression of $Neat1\_1$ displayed augmented activation of both NLRC4 and AIM2 inflammasomes in response to these stimuli (Fig. 2c, d). Moreover, activation of NLRC4 and AIM2 inflammasomes was attenuated in $Neat1^{-/-}$ BMDMs compared to $Neat1^{+/+}$ BMDMs (Fig. 2e, f).

In addition, overexpression of $Neat1\_1$ enhanced not only the interactions of ASC with pro-caspase-1 and NLRC4 in flagellin-activated iBMDMs (Fig. 2g), but also the interactions of ASC with pro-caspase-1 and AIM2 in poly(dA:dT)-activated iBMDMs (Fig. 2h). Therefore, $Neat1$ enhances the assembly and activation of diverse canonical inflammasomes that mediate the activation of caspase-1.

**Effect of $Neat1$ on NLRP3 inflammasome is IL-6-independent.** $Neat1$ could promote the expression of IL-6[37], a pro-inflammatory cytokine that stimulates immune response. However, in $IL-6^{-/-}$ and $IL-6^{+/+}$ BMDMs (Supplementary Fig. 3a) treated with LPS and nigericin, both caspase-1 activation and IL-1β secretion occurred to a comparable extent (Fig. 3a and Supplementary Fig. 3b). Moreover, siRNA-mediated knockdown of total $Neat1$ (Supplementary Fig. 3c, d), or $Neat1\_2$ alone (Supplementary Fig. 3e), similarly inhibited caspase-1 activation and IL-1β secretion in $IL-6^{-/-}$ and $IL-6^{+/+}$ BMDMs (Fig. 3b–d). Thus, IL-6 does not appear to mediate the function of $Neat1$ on the NLRP3 inflammasome.

**$Neat1$ interacts with caspase-1 p20 domain.** Given that $Neat1$ associates with, and promote the assembly of, the NLRP3 inflammasome in an IL-6-independent manner, we evaluated which component(s) of the NLRP3 inflammasome interacts with $Neat1$. NLRP3, ASC, and pro-caspase-1 were immunoprecipitated by their specific antibodies from lysates of murine iBMDMs at different stages of stimulation: no stimulation, primed with LPS

only, and primed with LPS followed by stimulation with nigericin. Total $Neat1$ and $Neat1\_2$ were found in pro-caspase-1 immunoprecipitates from cells of all three stages, but in NLRP3 and ASC immunoprecipitates only from cells that are activated by both LPS and nigericin (Fig. 4a–c and Supplementary Fig. 4a–c).

In a complementary approach, we pulled down $Neat1$ from cell lysates using biotinylated $Neat1$ antisense DNA probes, and analyzed the presence of NLRP3 inflammasome components in the pull-down samples. This again revealed that pro-caspase-1 bound to $Neat1$ in cells of all three stages, whereas ASC and NLRP3 bound to $Neat1$ only in cells treated with both LPS and nigericin (Fig. 4d and Supplementary Fig. 4d). Thus, $Neat1$ likely interacts with pro-caspase-1 constitutively and directly, and with NLRP3 and ASC inducibly and indirectly when these components are incorporated into the inflammasome together with pro-caspase-1.

Likewise, during the activation of the NLRC4 and AIM2 inflammasomes, $Neat1$ bound to pro-caspase-1 directly in cells of all stages: non-stimulated, LPS-primed, and LPS-primed followed by flagellin or poly(dA:dT) stimulation, whereas $Neat1$ bound to NLRC4/AIM2 and ASC only upon flagellin-dependent NLRC4 inflammasome activation or poly(dA:dT)-dependent AIM2 inflammasome activation (Supplementary Fig. 4e–h).

Upon activation, caspase-1 is processed to the mature subunits p20 and p10. We observed that more p20 than p10 subunits were pulled down by the $Neat1$ antisense DNA probes (Fig. 4d). This observation suggested that $Neat1$ may directly associate with the p20 subunit of the mature caspase-1 or the region corresponding to p20 on pro-caspase-1, and indirectly with the p10 subunit within the assembled mature caspase-1 hetero-tetramer. To verify this, we generated a panel of caspase-1 deletion mutations, each tagged with the Flag epitope (Supplementary Fig. 5a, b). These mutants, as well as full-length pro-caspase-1, were expressing in 293T cells, and the cell lysates were incubated with biotin-labeled $Neat1\_1$. This assay showed that $Neat1\_1$ bound to pro-caspase-1 and the p20 region, but not the caspase recruitment domain (CARD) or the p10 region (Fig. 4e). Furthermore, deletion of the p20 region (Δp20), but not the deletion of the CARD (ΔCARD) or the p10 region (Δp10), abolished the interaction with $Neat1\_1$ (Fig. 4f). Therefore, the p20 region of pro-caspase-1 is both sufficient and necessary for the interaction with $Neat1$, and this interaction persists upon caspase-1 activation.

To map the caspase-1-binding region on $Neat1$, we performed NLRP3 CLIP-seq peaks analysis. A strong and specific peak in the 5′ region of $Neat1\_1$ and $Neat1\_2$, along with some other weak peaks in the rest of $Neat1\_1$, was identified (Supplementary Fig. 5c), consistent with binding of total $Neat1$ and $Neat1\_2$ with the NLRP3 inflammasome. We generated three $Neat1\_1$ segments according to the CLIP-seq analysis (Supplementary Fig. 5c), and incubated each of them with HA-tagged pro-caspase-1. Only full-length $Neat1\_1$ and segment 1, which contained the 5′ 1 kb region, could pull down HA-tagged pro-caspase-1 (Supplementary Fig. 5d), suggesting that $Neat1$ binds with pro-caspase-1 via its 5′ region.

**$Neat1$ stabilizes the caspase-1 hetero-tetramers.** The active caspase-1, an $\alpha_2\beta_2$ hetero-tetramer formed by two p10 subunits and two p20 subunits[41], is turned over by the ubiquitin-proteasome pathway[42]. The hetero-tetramer is also unstable and readily dissociates[43]. $Neat1\_1$ did not affect the ubiquitination of either pro-caspase-1 or the p20 subunit (Supplementary Fig. 6a). Instead, $Neat1$ increased the interaction between p20 and p10 (Fig. 5a, b and Supplementary Fig. 6b, c), suggesting that $Neat1$ stabilizes the $(p20:p10)_2$ hetero-tetramer. Mature caspase-1 can also exist as a transient $(p33:p10)_2$ species, where p33 contains

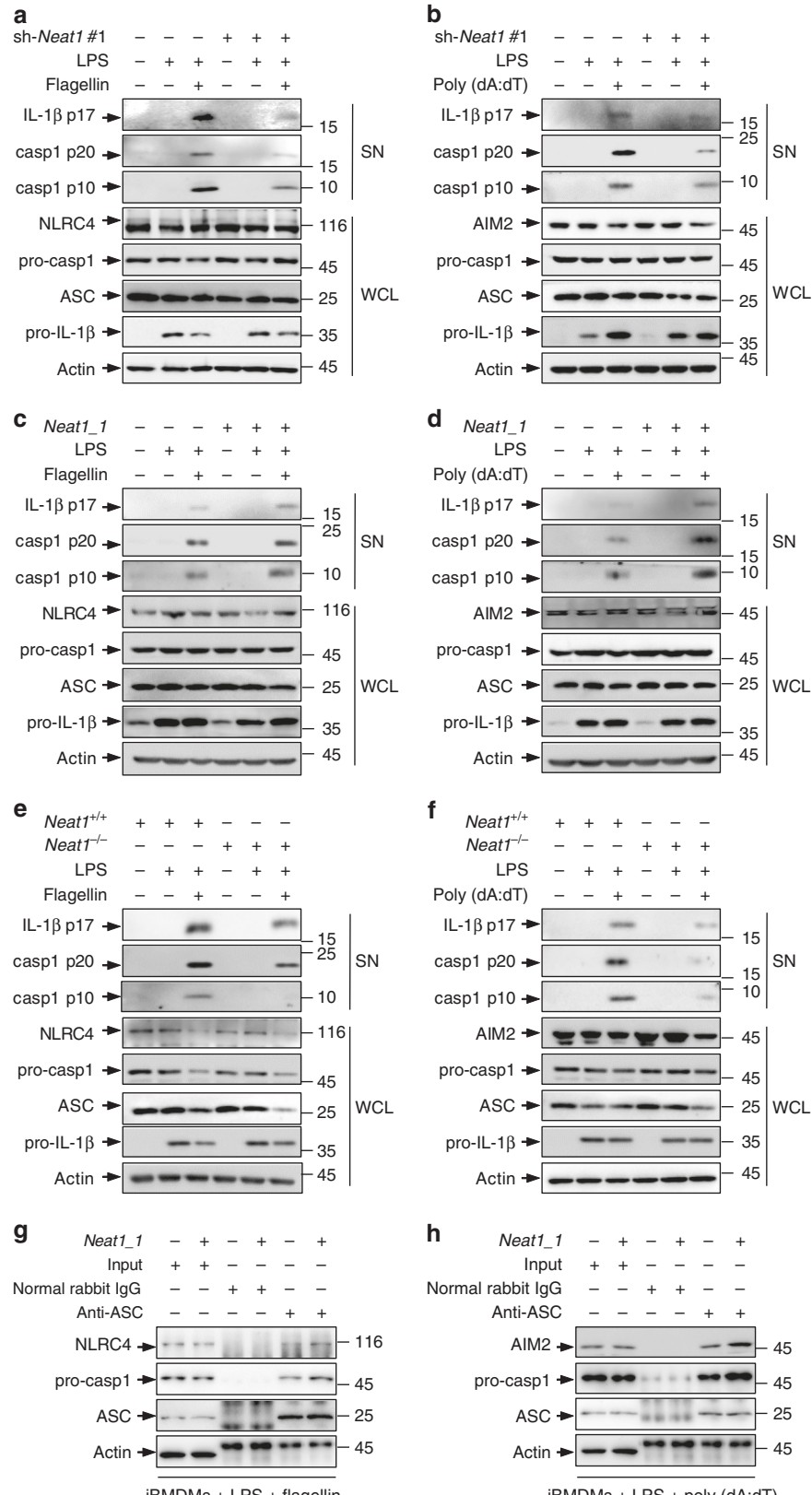

**Fig. 2** *Neat1* enhances the activation of NLRC4 and AIM2 inflammasomes. **a–d** iBMDMs devoid of *Neat1* by shRNA #1 (**a**, **b**), iBMDMs overexpressing *Neat1_1* (**c**, **d**), and the corresponding control iBMDMs were primed with LPS, and then stimulated with flagellin (**a**, **c**) or poly(dA:dT) (**b**, **d**). WCL and SN were analyzed by western blotting. **e**, **f** BMDMs from *Neat1*[+/+] or *Neat1*[−/−] mice were primed with LPS and then stimulated with flagellin (**e**) or poly(dA: dT) (**f**). WCL and SN were analyzed by western blotting. (**g**, **h**) Control iBMDMs and iBMDMs ectopically expressing *Neat1_1* were primed with LPS and then stimulated with flagellin (**g**) or poly(dA:dT) (**h**). Cell lysates were immunoprecipitated with normal rabbit IgG or ASC antibody. The samples were analyzed by western blotting as indicated. Source data are provided as a Source Data file

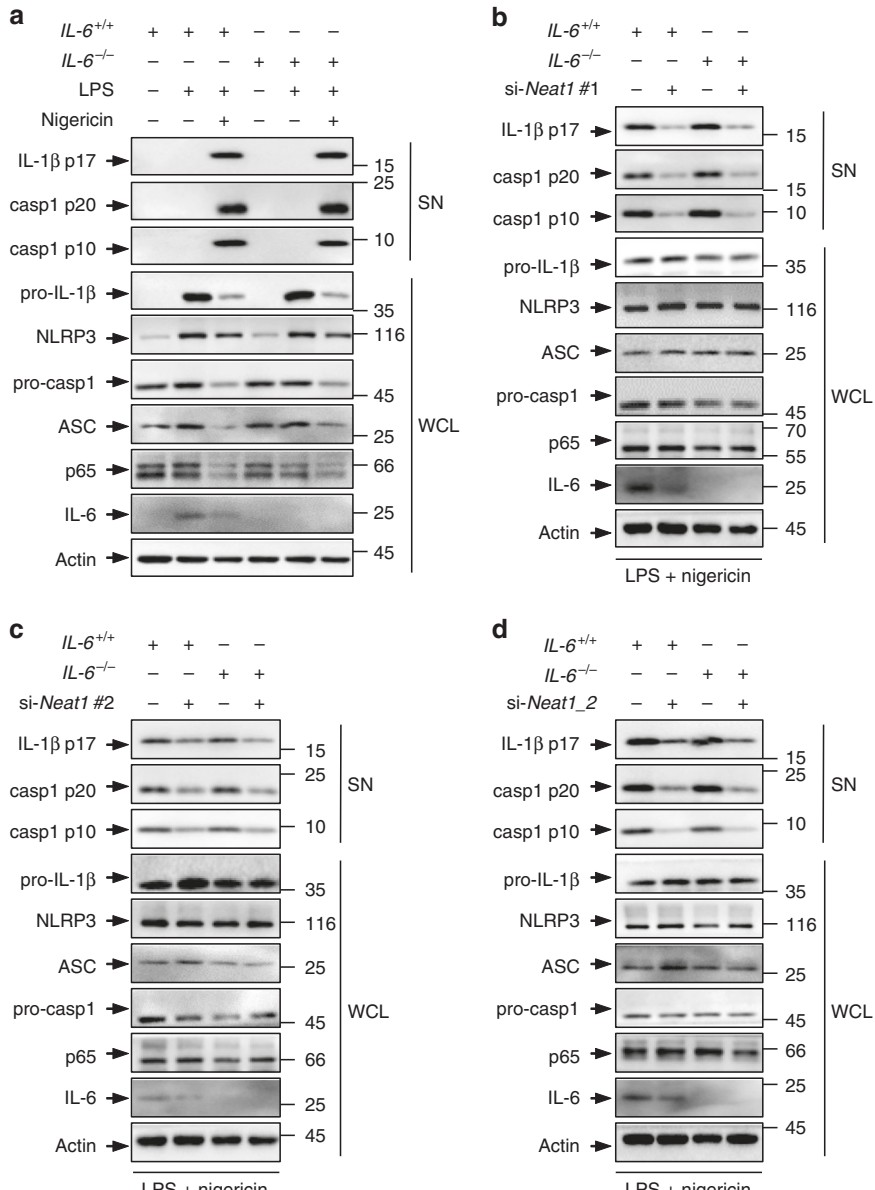

**Fig. 3** The effect of *Neat1* on the NLRP3 inflammasome activation is independent of IL-6. **a** BMDMs derived from *IL-6* wild-type (*IL-6*[+/+]) or knockout (*IL-6*[−/−]) mice were primed with LPS and then treated with nigericin. Protein levels in total cell lysates and supernatant fraction were analyzed by western blotting. **b–d** *IL-6*[+/+] or *IL-6*[−/−] BMDMs were transfected with control siRNA (**b–d**), *Neat1* siRNA #1 (**b**), *Neat1* siRNA #2 (**c**), or *Neat1_2* siRNA (**d**). Twenty-four hours later, cells were stimulated with LPS and nigericin. WCL and SN of above cells were subjected to western blotting with indicated antibodies. Source data are provided as a Source Data file

both the CARD and p20 regions[44]. Of note, *Neat1_1* could also increase the interaction between p33 and p10 (Fig. 5c, d and Supplementary Fig. 5d, e). Therefore, *Neat1* stabilizes the subunit–subunit interactions of mature caspase-1 hetero-tetramers. Consistently, caspase-1 activity was significantly reduced in *Neat1*[−/−] BMDMs compared to *Neat1*[+/+] BMDMs upon the formation of NLRP3, NLRC4, or AIM2 inflammasomes (Fig. 5e), while caspase-1 activity was strongly increased in *Neat1_1*-overexpressing cells compared to control cells upon the formation of these canonical inflammasomes (Fig. 5f). Therefore, *Neat1* strengthens the inter-subunit interactions within mature caspase-1 and augments the protease activity.

**Neat1 promotes caspase-1-dependent pyroptosis.** In addition to cytokine secretion, activation of caspase-1 by canonical inflammasomes results in pyroptosis[45]. We observed significant cell death among LPS and nigericin co-stimulated iBMDMs, but not untreated or LPS-primed iBMDMs, as shown by propidium iodide-positive staining and lactate dehydrogenase (LDH) release. Overexpression of *Neat1_1* noticeably augmented LPS/nigericin-induced pyroptosis (by ~120%; Fig. 6a, b), whereas *Neat1* silencing dramatically reduced pyroptosis (by ~80%; Fig. 6c, d). Likewise, *Neat1* knockout rendered BMDMs high resistant to pyroptosis (reduced by ~70%; Fig. 6e, f). Therefore, *Neat1* promotes caspase-1-dependent pyroptotic cell death.

**Neat1 is exported to cytoplasm upon inflammasome activation.** Murine *Neat1* and its human ortholog are highly abundant lncRNAs that are mainly localized to the nucleus of unstressed cells[46], while assembly of various canonical inflammasomes occurs in the cytoplasm[47,48]. We found that, upon the stimulation of LPS and nigericin, the levels of total *Neat1* or the *Neat1_2*

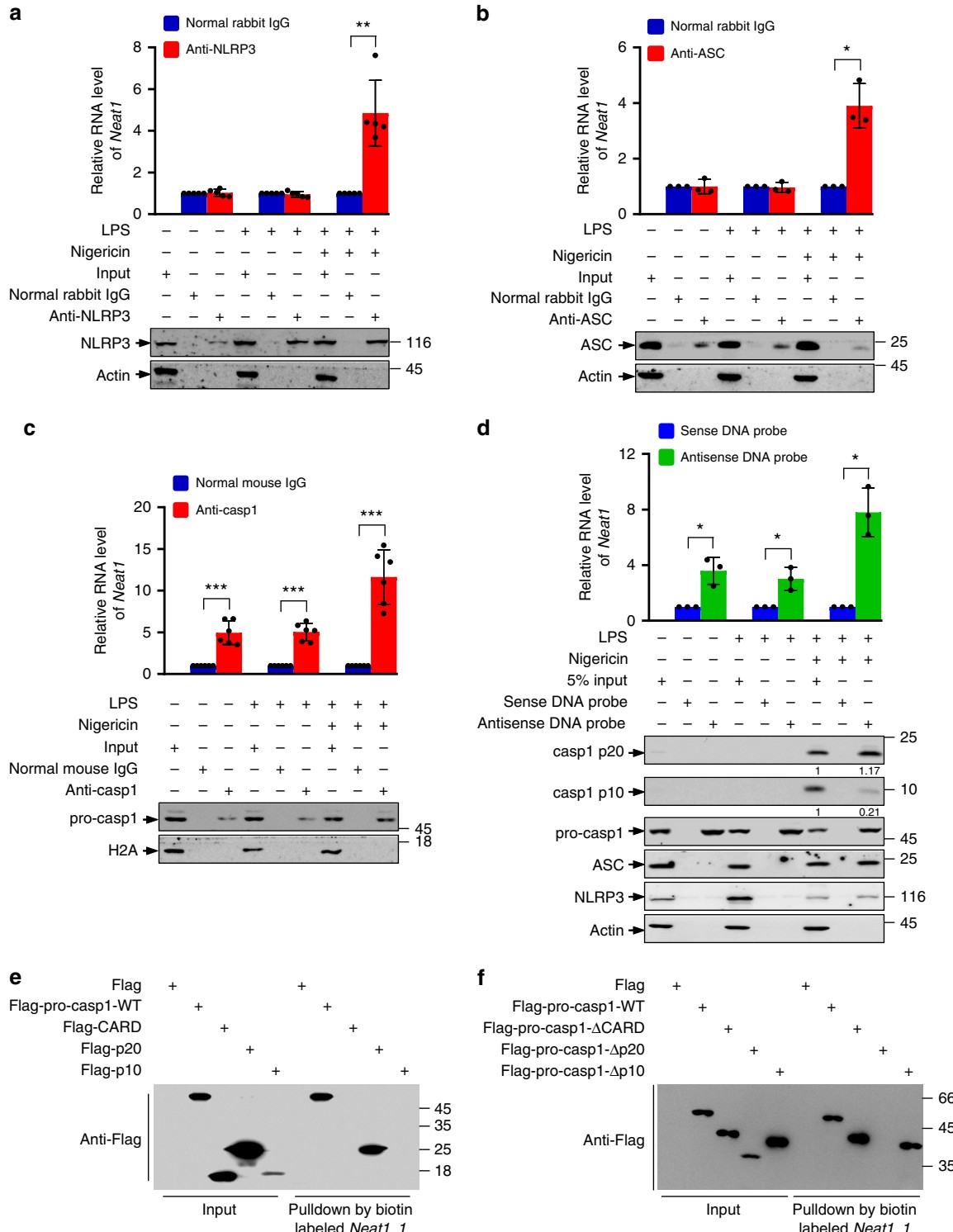

**Fig. 4** *Neat1* interacts with caspase-1 p20 domain. **a**–**c** Cell lysates of normal iBMDMs, LPS-primed iBMDMs or LPS-nigericin-co-stimulated iBMDMs were incubated with normal rabbit IgG (**a**–**c**), NLRP3 antibody (**a**), ASC antibody (**b**), or caspase-1 antibody (**c**) for RIP. The immunoprecipitates were analyzed by real-time RT-PCR to exam enrichment efficiency of *Neat1* and by Western blotting for protein levels. **d** Cell lysates of normal, LPS-primed, or LPS-nigericin-co-stimulated iBMDMs were incubated with biotin-labeled *Neat1* sense or antisense DNA probes immobilized on beads. The precipitated samples were analyzed by real-time RT-PCR for *Neat1* contents, and by western blotting for indicated protein levels. **e** In vitro transcribed biotin-labeled *Neat1_1* was incubated with lysates extracted from 293T cells expressing Flag-tagged wild-type pro-caspase-1, CARD, p20, or p10 for 3 h. The input and biotin pull-down samples were then carried out to western blotting. **f** In vitro transcribed biotin-labeled *Neat1_1* was incubated with lysates extracted from 293T cells expressing Flag-tagged wild-type pro-caspase-1 or the indicated deletion mutants for 3 h. The input and biotin pull-down samples were analyzed by western blotting. In **a**, data shown are mean ± SD ($n = 5$). In **b** and **d**, data shown are mean ± SD ($n = 3$). In **c**, data shown are mean ± SD ($n = 6$). *$P < 0.05$, **$P < 0.01$, two-tailed *t*-test. Source data are provided as a Source Data file

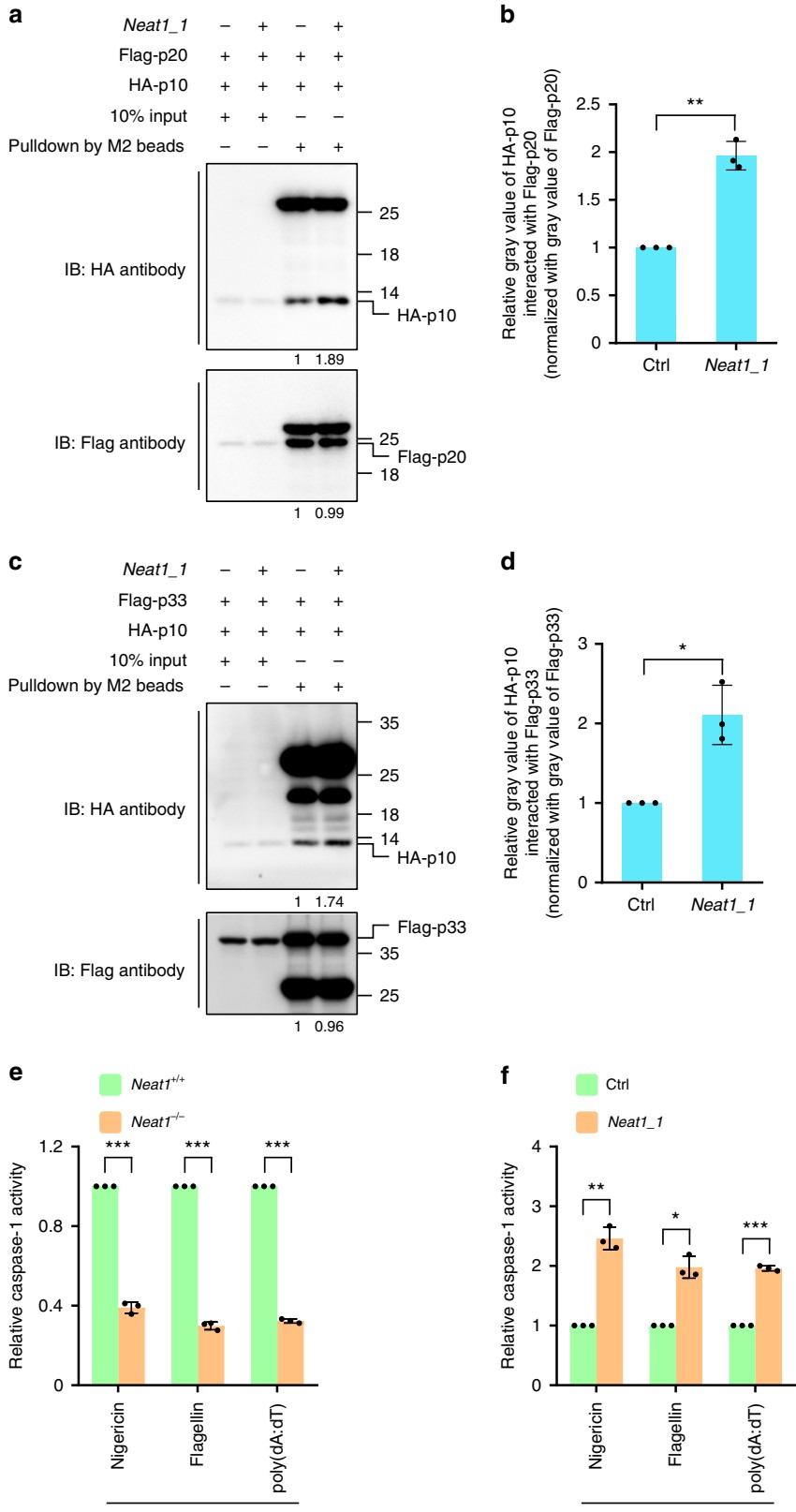

**Fig. 5** *Neat1* stabilizes the caspase-1 hetero-tetramers and increases caspase-1 activity. **a–d** 293T cells were infected with lentiviruses expressing control RNA or *Neat1_1* and then co-transfected with Flag-p20 and HA-p10 (**a**, **b**) or Flag-p33 and HA-p10 (**c**, **d**). Cell lysates were incubated with anti-Flag mAb (M2) beads. Input samples and pull-down products were subjected to western blotting (**a**, **c**). Relative ratios of HA-p10 pulled down by Flag-p20 (**b**) or Flag-p33 (**d**) were shown in histogram, based on the data of three independent replication experiments. **e** *Neat1*$^{+/+}$ and *Neat1*$^{-/-}$ BMDMs were primed with LPS and then activated with nigericin, flagellin, or poly(dA:dT). Mature caspase-1 activity was measured. **f** Control or *Neat1_1* overexpressing iBMDMs were primed with LPS and then activated with nigericin, flagellin, or poly(dA:dT). Mature caspase-1 activity were measured. In **b**, **d**, **e** and **f**, data shown are mean ± SD ($n = 3$). *$P < 0.05$, **$P < 0.01$, ***$P < 0.001$, two-tailed $t$-test. Source data are provided as a Source Data file

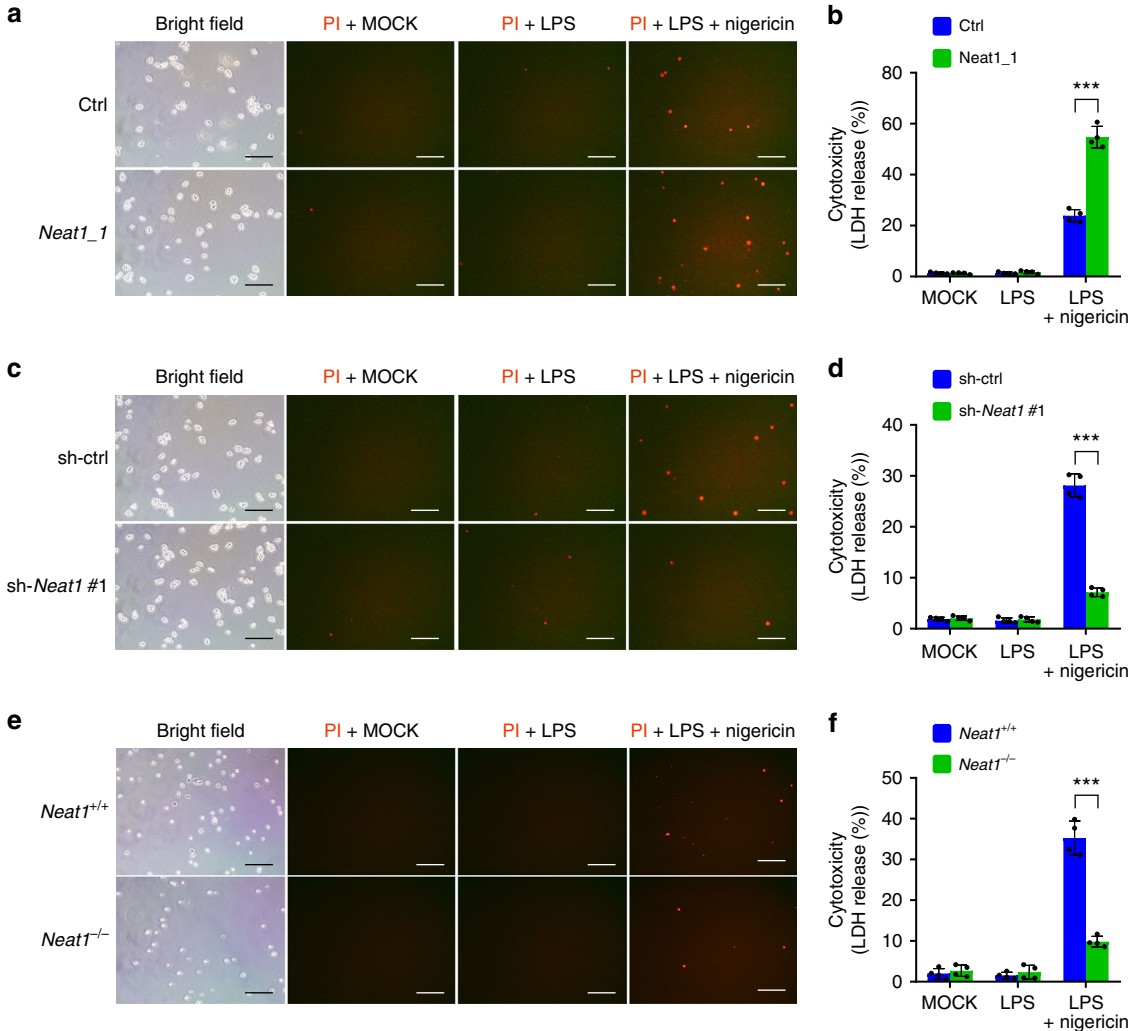

**Fig. 6** *Neat1* promotes caspase-1-dependent pyroptosis. **a–f** *Neat1_1*-overexpressing iBMDMs (**a**, **b**), *Neat1*-knockdown iBMDMs (**c**, **d**), *Neat1*$^{-/-}$ BMDMs (**e**, **f**), and the corresponding control cells were primed with LPS and then stimulated with nigericin. Cells were stained with PI and analyzed under microscope (**a**, **c**, and **e**). Percentages of pyroptotic cells were measured by LDH release into supernatants (**b**, **d**, and **f**). In **a**, **c**, and **e**, the bar graph indicates 300 μm. In **b**, **d**, and **f**, data shown are mean ± SD ($n = 4$). ***$P < 0.001$, two-tailed $t$-test. Source data are provided as a Source Data file

isoform remained unchanged (Supplementary Fig. 7a). To understand how *Neat1* participates in the inflammasome activation, we examined the presence of *Neat1* in the cytoplasmic fraction and culture medium of cells treated under different conditions. Compared with untreated or LPS-primed iBMDMs, LPS and nigericin co-stimulated iBMDMs contained more *Neat1* in the cytoplasm (~8.5% versus ~1%; Fig. 7a and Supplementary Fig. 7b). Moreover, consistent with the increased pyroptosis, LPS and nigericin co-stimulated iBMDMs also released more *Neat1* into the culture medium (Supplementary Fig. 7c). Similarly, more *Neat1* was released from nucleus to cytoplasm in LPS/flagellin- or LPS/poly(dA:dT)-treated iBMDMs compared to untreated or LPS-primed iBMDMs (Fig. 7b, c and Supplementary Fig. 7d, e). Additionally, we treated iBMDMs with LPS and 2DG to stimulate the formation of Nlrp1b inflammasome. This also led to the translocation of *Neat1* to the cytoplasm (Supplementary Fig. 7f, g). These observations suggest that *Neat1* is released from nucleus to the cytoplasm during the activation of the NLRP3, NLRC4, AIM2, or Nlrp1b inflammasome.

To corroborate the observations, we performed RNA fluorescence in situ hybridization (RNA-FISH) to assess the cellular localization of *Neat1*. Among untreated or LPS-primed iBMDMs, *Neat1* remained in the nucleus while ASC was diffusely localized in the cytoplasm (Fig. 7d). However, in LPS and nigericin co-stimulated iBMDMs where ASC became polymerized, a portion of *Neat1* was released to the cytoplasm and became colocalized with ASC (Fig. 7d), further supporting the notion that the nuclear export of *Neat1* is induced in activated cells to facilitate inflammasomes formation.

**Perturbation of paraspeckles leads to the release of *Neat1*.** *Neat1* is concentrated in paraspeckles and is an essential architectural component of these nuclear bodies[35,36]. To examine how LPS and nigericin co-stimulation may induce *Neat1* nuclear export, we examined levels of three main paraspeckle proteins, Nono, Pspc1, and Sfpq[35]. Compared with control or LPS-treated iBMDMs, Pspc1 and the short isoform of Sfpq (Sfpq-S) declined in their abundance in LPS and nigericin co-treated iBMDMs, while the levels of Nono and the long isoform of Sfpq (Sfpq-L) remained unchanged (Fig. 8a). Levels of Pspc1 and Sfpq-S were also decreased in iBMDMs activated by flagellin or poly(dA:dT) (Fig. 8b, c). In addition, levels of Pspc1 declined in 2DG-activated iBMDMs (Supplementary Fig. 8a). Accordingly, the size and number of paraspeckles were dramatically reduced in LPS and nigericin co-stimulated iBMDMS compared to untreated or

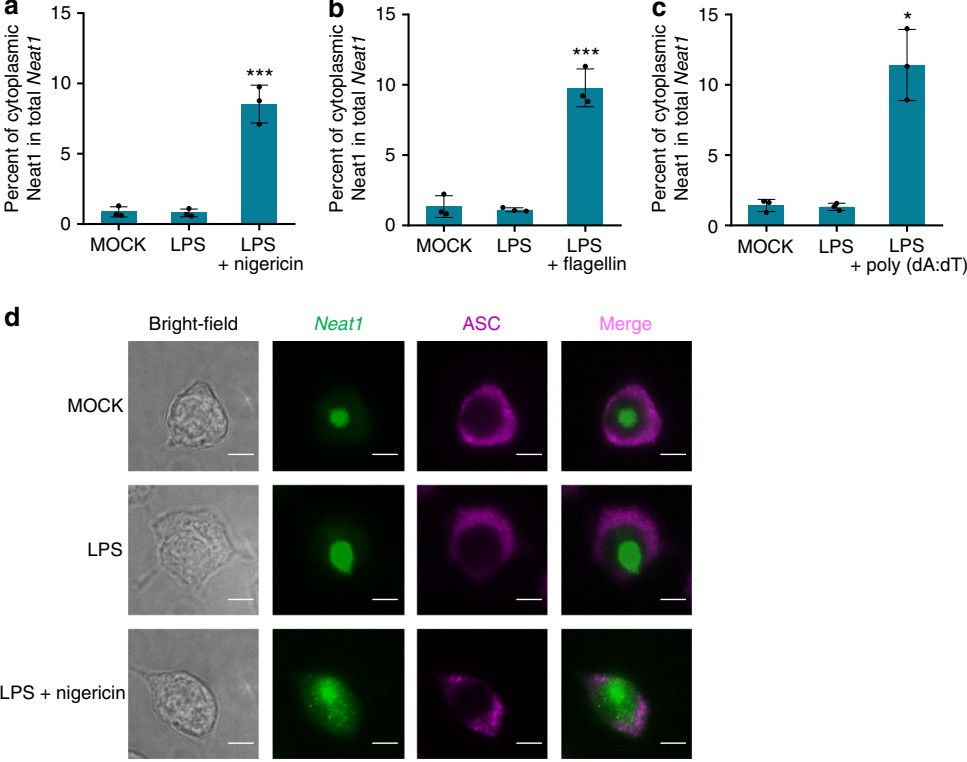

**Fig. 7** *Neat1* is exported from nucleus to cytoplasm in response to inflammasome-activating stimuli. **a–c** Untreated iBMDMs (**a–c**), LPS-primed iBMDMs (**a–c**), nigericin-activated iBMDMs (**a**), flagellin-activated iBMDMs (**b**), or poly(dA:dT)-activated iBMDMs (**c**) were subjected to fractionation into cytoplasmic and nuclear extracts. Cytoplasmic RNA was analyzed for the *Neat1* expression by real-time RT-PCR. **d** The subcellular co-localization of *Neat1* (green) and ASC (purple) in untreated, LPS-primed or LPS-nigericin-co-stimulated iBMDMs was analyzed by RNA-FISH and IF. In **d**, the bar graph indicates 5 μm. In **a**, **b**, and **c**, data shown are mean ± SD ($n = 3$). $*P < 0.05$, $***P < 0.001$, two-tailed $t$-test. Source data are provided as a Source Data file

LPS-primed iBMDMs (Fig. 8d and Supplementary Fig. 8b, c). Consistently, the interaction of Nono with Pspc1 and Sfpq was reduced in response to the activation of the NLRC4, AIM2, and Nlrp1b inflammasome (Fig. 8e, f and Supplementary Fig. 8d). All above results claim that activation of various inflammasomes weakens assembly of paraspeckles, which store *Neat1* in nucleus.

To verify that the perturbation of paraspeckles results in the release of *Neat1* into the cytoplasm, we knocked down Nono, Pspc1, and Sfpq individually (Supplementary Fig. 8e, f). In each case, the nuclear export of both total *Neat1* and *Neat_2* was observed, and a substantial fraction of *Neat1* (~15–20%) was found in the cytoplasm (Fig. 8g, h). Therefore, stimuli, that activate various inflammasomes, disrupt paraspeckles leading to the translocation of *Neat1* to the cytoplasm, where *Neat1* influences the assembly of inflammasomes.

**Neat1 mediates the effect of HIF-2α on inflammasomes**. The expression of *Neat1* is induced by infection of viruses (e.g., influenza virus and herpes simplex virus)[49,50], and is also regulated by the tumor suppressor p53[51], a central sentinel for various internal and external stresses[52,53]. In addition, hypoxia, which promotes inflammation via HIF-1α-mediated induction of NLRP3 and IL-1β[24,25], can induce *Neat1* through HIF-2α-mediated transcriptional activation[54,55]. Thus, *Neat1* may function as a stress sensor for associated inflammasomes. To test this possibility, we investigated the role of *Neat1* in hypoxia-mediated activation of the NLRP3 inflammasome. iBMDMs cultured under hypoxic conditions exhibited increased levels of *Neat1* (Supplementary Fig. 9a), HIF-1α, HIF-2α, p65, NLRP3, and IL-1β (Fig. 9a), which was accompanied by enhanced NLRP3 inflammasome formation, caspase-1 activation, and IL-1β secretion

(Fig. 9a). However, knockout of *Neat1* not only reduced the activation of NLRP3 inflammasome under normoxic conditions, but also effectively prevented the increase of inflammasome activation under hypoxic conditions (Supplementary Fig. 9b), underscoring a critical role for *Neat1* in hypoxia-induced NLRP3 inflammasome activation. In addition, activation of NLRC4 and AIM2 inflammasomes was also increased under hypoxic conditions, along with increased expression of HIF-1α, HIF-2α, and IL-1β (Fig. 9b, c). Knocking down either HIF-1α (Fig. 9d, f, h) or HIF-2α (Fig. 9e, g, i) reduced hypoxia-induced caspase-1 activation and IL-1β maturation under the condition of NLRP3, NLRC4, or AIM2 inflammasome activation (Fig. 9d–i), indicating that both HIF-1α and HIF-2α mediate the effect of hypoxia on canonical inflammasomes. HIF-1α knockdown, but not HIF-2α knockdown, reduced the activation of diverse inflammasomes in iBMDMs devoid of *Neat1* (Fig. 9d–i), which claims that Neat1-silencing could efficiently block the effect of HIF-2α, but not HIF-1α, on inflammasomes activation. These results suggest that hypoxia-induced activation of the NLRP3, NLRC4, and AIM2 inflammasomes and associated inflammatory responses are in part due to HIF-2α-mediated upregulation of *Neat1*.

**Loss of Neat1 reduces inflammation in vivo**. To investigate the effect of *Neat1* in vivo, we used an NLRP3 inflammasome-mediated, IL-1-dependent mouse peritonitis model induced by the aluminum salt alum[56]. We intraperitoneal (i.p.) injected alum in *Neat1*[+/+] and *Neat1*[−/−] mice to induce peritonitis, and assayed IL-1β in the lavage fluid. A significant reduction of IL-1β secretion in the lavage fluid was observed in *Neat1*[−/−] mice compared to *Neat1*[+/+] mice (by ~67%; Fig. 10a). Moreover, collected peritoneal exudate cells (PECs) and analyzed alum-induced recruitment of

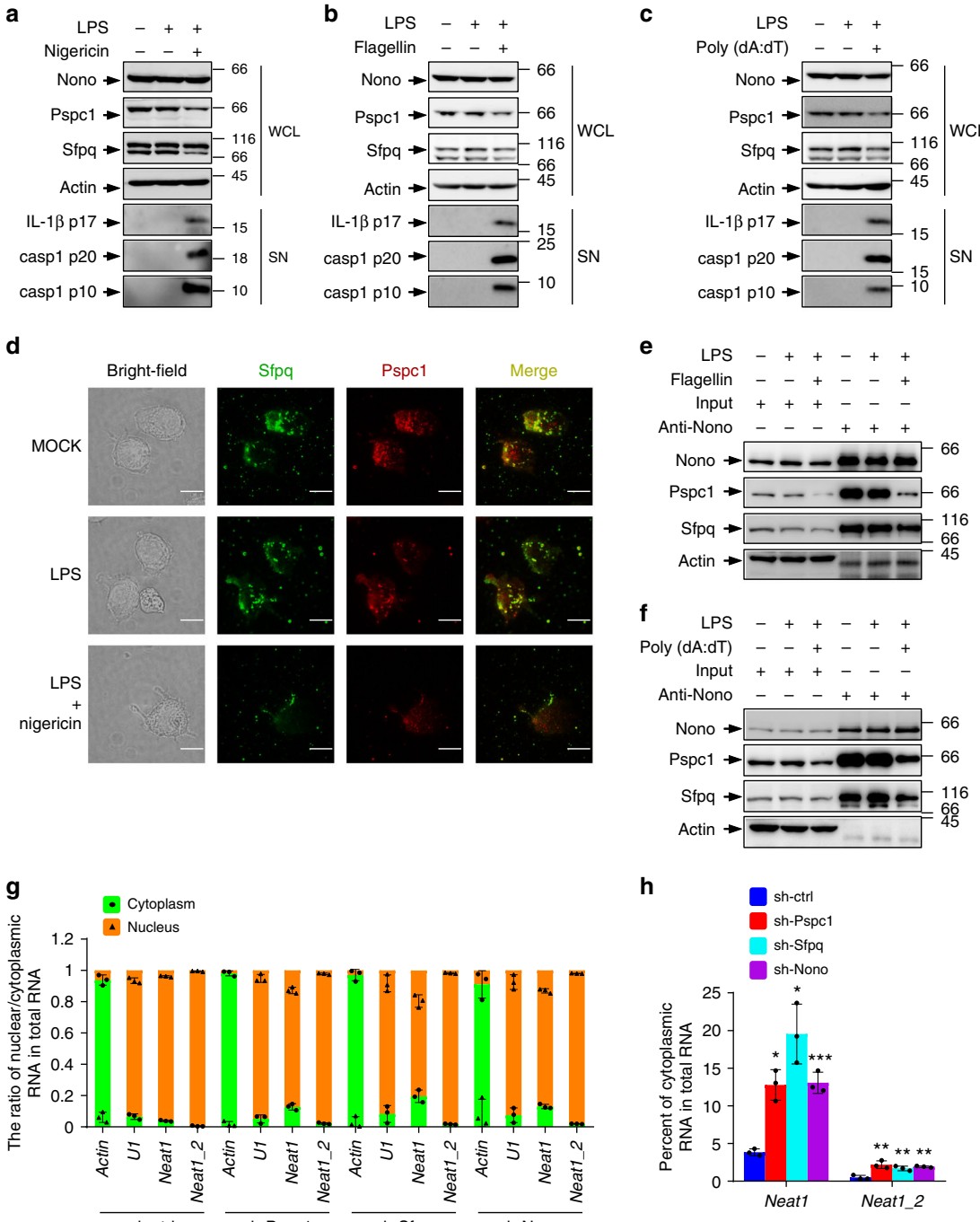

**Fig. 8** *Neat1* is released from nuclear paraspeckles into cytoplasm during inflammasome activation. **a–c** Cell lysates of untreated iBMDMs (**a–c**), LPS-primed iBMDMs (**a–c**), nigericin-activated iBMDMs (**a**), flagellin-activated iBMDMs (**b**), or poly(dA:dT)-activated iBMDMs (**c**) were analyzed by Western blotting. **d** Colocation of paraspeckle proteins, Sfpq (green), and Pspc1 (red) were assayed by immunostaining. The location of paraspeckles (yellow) was obtained by merging two signals of Sfpq and Pspc1. The position of the nucleus is shown in the bright-field image. **e, f** Interactions among paraspeckle components in untreated iBMDMs (**e, f**), LPS-primed iBMDMs (**e, f**), flagellin-activated iBMDMs (**e**), or poly(dA:dT)-activated iBMDMs (**f**) were analyzed by co-immunoprecipitation assay. **g, h** iBMDMs infected with lentiviruses expressing control, Pspc1, Sfpq, or Nono shRNA were analyzed for the cytoplasmic and nuclear levels of *Neat1* and *Neat1_2*. Percentages of cytoplasmic *Neat1* or *Neat1_2* among total *Neat1* or *Neat1_2* RNA were shown in **h**. In **d**, the bar graph indicates 10 μm. In **g** and **h**, data shown are mean ± SD ($n = 3$). *$P < 0.05$, **$P < 0.01$, ***$P < 0.001$, two-tailed *t*-test. Source data are provided as a Source Data file

inflammatory cells by flow cytometry (Supplementary Fig. 10a). The number of total PECs recruited upon alum challenge was markedly decreased by 58% in *Neat1*$^{-/-}$ mice (Fig. 10b). Alum-induced recruitment of neutrophils and Ly6C$^+$ monocytes was also strongly inhibited in the *Neat1*$^{-/-}$ mice (Fig. 10c, d).

To evaluate the role of *Neat1* on NLRC4 inflammasome activation in vivo, we used a mouse model of flagellin-induced pneumonia. *Neat1*$^{+/+}$ and *Neat1*$^{-/-}$ mice were intranasally instilled with TurboFect reagent and flagellin to induce pneumonia. We found that IL-1β secretion in bronchoalveolar

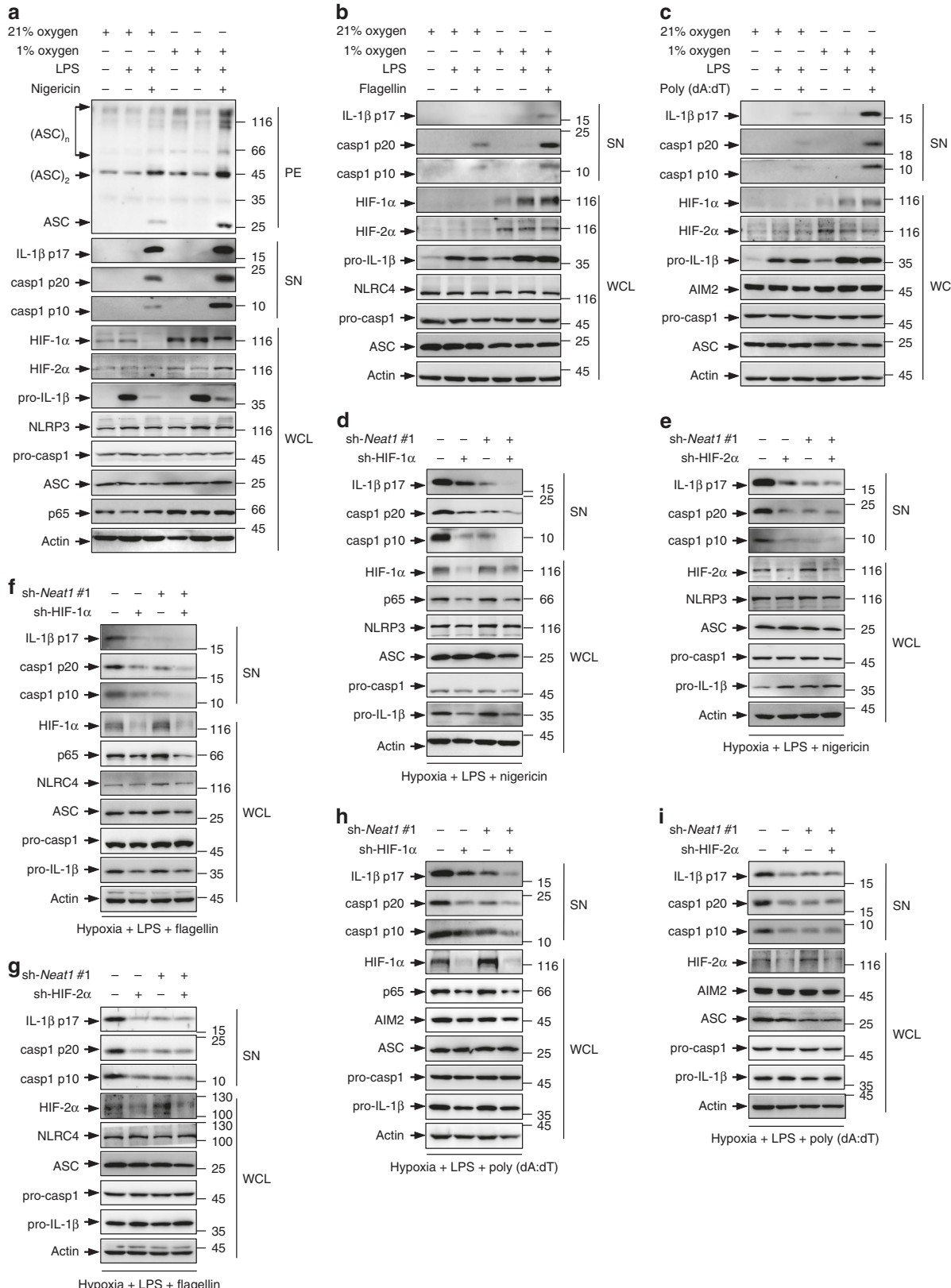

**Fig. 9** *Neat1* mediates the effect of HIF-2α on inflammasomes under hypoxic condition. **a–c** iBMDMs cultured under normoxic or hypoxic conditions were untreated (**a–c**), primed with LPS (**a–c**, or co-stimulated with LPS and nigericin (**a**), flagellin (**b**), or poly(dA:dT) (**c**). WCL (**a–c**), SN (**a–c**), and PE (**a**) were subjected to western blotting. **d–i** Control (**d–i**), HIF-1α-silenced (**d**, **f**, and **h**), HIF-2α-silenced (**e**, **g** and **i**), *Neat1*-silenced (**d–i**), HIF-1α-*Neat1*-double-silenced (**d**, **f** and **h**), or HIF-2α-*Neat1*-double-silenced (**e**, **g** and **i**) iBMDMs were cultured under hypoxic conditions and primed with LPS, followed by stimulation with nigericin (**d**, **e**), flagellin (**f**, **g**), or poly(dA:dT) (**h**, **i**). WCL and SN were analyzed by western blotting. Source data are provided as a Source Data file

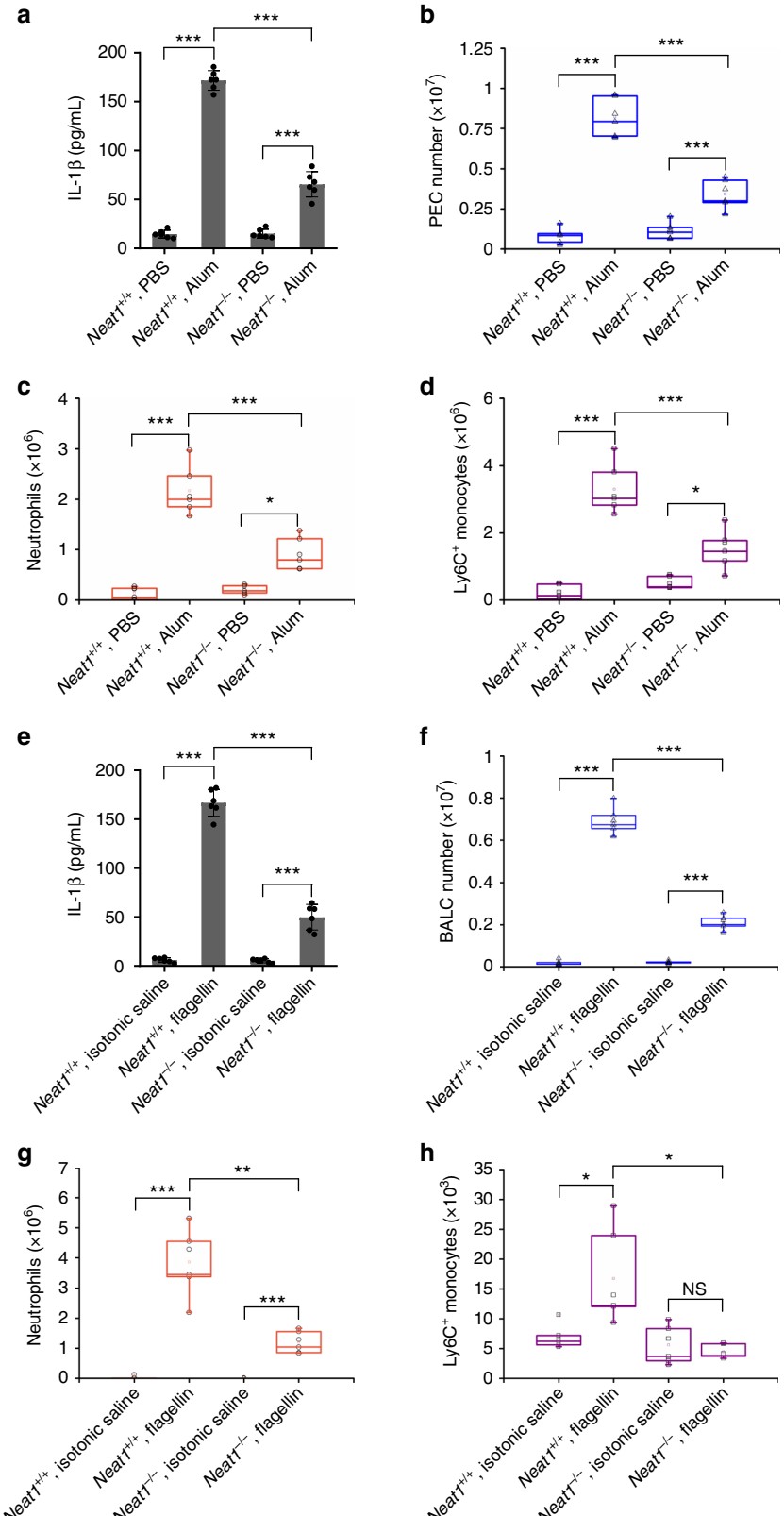

**Fig. 10** Loss of *Neat1* impairs alum-induced peritoneal inflammation and attenuates flagellin-induced lung inflammation in vivo. **a** IL-1β content in the lavage fluid from *Neat1*[+/+] or *Neat1*[−/−] mice 12 h after PBS or alum injection. **b** Absolute numbers of *Neat1*[+/+] or *Neat1*[−/−] PECs recovered 12 h after PBS or alum injection. **c**, **d** Absolute numbers of neutrophils (**c**) and Ly6C[+] monocytes (**d**) exuded to the peritoneum 12 h after PBS or alum injection. **e** IL-1β content in the BALF from *Neat1*[+/+] or *Neat1*[−/−] mice 12 h after isotonic saline or flagellin intranasal instillation. **f** Absolute numbers of *Neat1*[+/+] or *Neat1*[−/−] PECs recovered 12 h after isotonic saline or flagellin intranasal instillation. **g**, **h** Absolute numbers of neutrophils (**g**) and Ly6C[+] monocytes (**h**) exuded to the lung 12 h after isotonic saline or flagellin intranasal instillation. In **a**–**h**, data shown are mean ± SD (*n* = 6). *P < 0.05, **P < 0.01, ***P < 0.001, two-tailed *t*-test. Source data are provided as a Source Data file

lavage fluid (BALF) was decreased by 70% in flagellin-challenged $Neat1^{-/-}$ mice compared to flagellin-challenged $Neat1^{+/+}$ mice (Fig. 10e). We also analyzed bronchoalveolar lavage cells (BALCs) and flagellin-induced recruitment of inflammatory cells by flow cytometry (Supplementary Fig. 10b). The total number of BALCs recruited upon flagellin challenge was decreased by 70% in $Neat1^{-/-}$ mice (Fig. 10f). Furthermore, flagellin-induced recruitment of neutrophils and $Ly6C^+$ monocytes was strongly decreased in $Neat1^{-/-}$ mice (Fig. 10g, h).

The overt defects of $Neat1^{-/-}$ mice in alum-induced peritoneal inflammation and flagellin-induced lung inflammation emphasize an important function for $Neat1$ in the activations of the canonical inflammasomes, the subsequent accumulation of immune cells and the associated symptoms in vivo.

## Discussion

Proper control of inflammasome assembly and caspase-1 activation permits effective antimicrobial and inflammatory responses while avoiding tissue damage. Yet, how these supramolecular assemblies are regulation remains incompletely defined. LncRNAs are increasingly recognized as important regulators in different biological settings, but few lncRNAs are implicated in innate immunity[57–60]. The current work reveals that the lncRNA $Neat1$ plays a critical role in regulating various caspase-1-associated canonical inflammasomes including NLRC4, AIM2, and especially NLRP3 inflammasomes. $Neat1$ promotes the assembly of inflammasomes, and also stabilizes the mature caspase-1 tetramers, $(p20:p10)_2$, and $(p33:p10)_2$, and increases their protease activity. Thus, $Neat1$ is likely a previously unanticipated RNA component that enables the assembly of these largely proteinaceous signaling platforms to facilitate the recruitment, maturation, and stabilization of caspase-1 in activated macrophages. The ability of $Neat1$ to directly regulate inflammasomes provides a notable example for the role of lncRNAs in regulating large multimeric protein complexes in the cytoplasm.

The assembly of inflammasomes requires a common priming event, such as LPS-induced activation of TLRs, which induces the expression of NLRP3 and pro-IL-1β[2,5]. However, the NLRC4, AIM2, and especially NLRP3 inflammasomes are versatile inflammasome, collectively responding to a wide range of stimuli[5,24,61–64]. It remains unclear whether these stimuli also employ a general mechanism to modulate inflammasomes. Of note, the expression of $Neat1$ is highly regulated by various signals, including those that activates the tumor suppressor p53 (refs. [40,51]). Moreover, hypoxia is a key physiological stressor present within both normal and tumor microenvironments[65] and a critical determinant of the immune response. Here we show that $Neat1$ links the effect of hypoxia with the activation of NLRP3, NLRC4, and AIM2 inflammasomes. Both human $NEAT1$ and murine $Neat1$ are thought to maintain the structural integrity of the paraspeckles[35], the function of which remains unclear. We find that the paraspeckles are disrupted in response to various inflammasome-activating signals, leading to the release of $Neat1$ and subsequent translocation to the cytoplasm, implying that paraspeckles may serve as a storage depot for $Neat1$. Therefore, the upregulation and translocation of $Neat1$ may be a converging event leading to the activation of the various caspase-1-associated canonical inflammasomes by a wide range of stimuli, underscoring a common mechanism for the assembly of these inflammation signaling platforms. In addition, our finding, along with previous observation that ASC and NLRP3 are recruited from other cellular components to mitochondria and associated endoplasmic reticulum membranes upon both priming and activation processes[66], also highlights the trafficking of inflammasome components as a major mechanism for the regulation of inflammasome activity.

Finally, by generating $Neat1$ knockout mice and analyzing their response to alum-induced peritoneal inflammation and flagellin-induced lung inflammation, we demonstrate that $Neat1$ enhances the activation of NLRP3 and NLRC4 inflammasomes and promotes pyroptosis in vivo (Supplementary Fig. 11). Therefore, $Neat1$ may represent a therapeutic target for inflammasome-associated diseases such as gout and autoinflammatory syndromes.

## Methods

**Antibodies and reagents**. Antibodies used for western bloting included anti-IL-1β (R&D Systems, 1:1000, AF-401-NA), anti-HIF-2α (R&D Systems, 1:1000, AF2997), anti-NLRP3 (Adipogen, 1:1000, AG-20B-0014), anti-ASC (Adipogen, 1:1000, AG-25B-0006), anti-caspase-1 (p20) (Adipogen, 1:1000, AG-20B-0042), anti-caspase-1 (p10) (Adipogen, 1:1000, AG-20B-0044), anti-p65 (Cell Signaling Technology, 1:1000, 8242), anti-HIF-1α (Cell Signaling Technology, 1:1000, 14179), anti-AIM2 (Cell Signaling Technology, 1:1000, 13095), anti-IL-6 (Cell Signaling Technology, 1:1000, 12912), anti-Actin (CMC-TAG, 1:2000, AT0009), anti-FLAG (Sigma-Aldrich, 1:2000, F3165), anti-Pspc1 (Santa Cruz, 1:200, sc-374181), anti-Sfpq (Santa Cruz, 1:200, sc-271796), anti-Nono (ABclonal, 1:500, A5282), anti-NLRC4 (ABclonal, 1:500, A7382), anti-histone H2A (abcam, 1:1000, ab177308), HRP-linked anti-rabbit IgG (Cell Signaling Technology, 1:2000, 7074), HRP-linked anti-mouse IgG (Cell Signaling Technology, 1:2000, 7076), HRP-linked anti-goat IgG (Santa Cruz, 1:1000, sc-2354). Antibodies used for immunoprecipitation included anti-NLRP3 (Cell Signaling Technology, 1:200, 15101), anti-caspase-1 (p20) (Adipogen, 1:200, AG-20B-0042), anti-ASC (Adipogen, 1:200, AG-25B-0006), anti-Nono (ABclonal, 1:50, A5282), normal rabbit IgG (Cell Signaling Technology, 1:200, 2729), normal mouse IgG (Santa Cruz, 1:80, sc-2025). Anti-ASC (Santa Cruz, 1:50, sc-22514), anti-Pspc1 (mouse IgG, Santa Cruz, 1:50, sc-374181), anti-Sfpq (mouse IgG, Santa Cruz, 1:50, sc-271796), anti-Sfpq (rabbit IgG, Proteintech, 1:50, 15585-1-AP), anti-Nono (rabbit IgG, ABclonal, 1:50, A5282), Alexa fluor 488 donkey anti-rabbit IgG(H + L) (Life Technologies, 1:1000, A21206), Alexa fluor 568 donkey anti-mouse IgG(H + L) (Life Technologies, 1:1000, A10037), and Alexa fluor 647 donkey anti-rabbit IgG(H + L) (Life Technologies, 1:1000, A31573) were used for immunofluorescence. PerCP-Cy™5.5 anti-mouse CD11b (BD Biosciences, 1:1000, 550993), APC anti-mouse Ly-6G (BD Biosciences, 1:1000, 560599), and PE anti-mouse Ly-6C (BD Biosciences, 1:1000, 560592) were used for flow cytometry. LPS, nigericin, flagellin, and poly(dA:dT) were purchased from Invivogen; Lipofectamine 2000 Transfection Reagent was from Invitrogen; DOTAP Liposomal Transfection Reagent was from Roche; puromycin, MG132, propidium iodide, and 2-deoxy-D-Glucose were from Solarbio; proteinase K and RiboLock RNase Inhibitor were from ThermoFisher Scientific; and Igepal CA-630 was from Sigma. Caspase-1 Activity Assay Kit purchased from Beoytime was used to perform the semi-quantitative analysis of mature caspase-1 activity.

**Mice**. $Neat1^{-/-}$ mice on a C57BL/6 background were generated by Biocytogen Biological Technology Co., Ltd. Briefly, C57BL/6 mice and Kunming (KM) mice were housed in a specific pathogen free facility. For $Neat1$ targeting, two sgRNAs (Supplementary Table 1) were designed to target either a region upstream or downstream of the gene $Neat1$. For each targeting site, candidate guide RNAs were designed by the CRISPR design tool (http://crispr.mit.edu). Guide RNAs were screened for on-target activity use UCATM. UCATM (Universal CRISPR Activity Assay), a sgRNA activity detection system developed by Biocytogen, is simpler and more sensitive than MSDase assay. The Cas9 mRNA and sgRNAs were transcribed by T7 RNA polymerase in vitro. For Cas9 mRNA and sgRNAs production, add the T7 promoter sequence to the Cas9 or sgRNA template by PCR amplification. The T7-Cas/sgRNA PCR products was gel purified and used as the template for in vitro transcription using the MEGA shortscript T7 kit (Life Technologies) according to the kit protocol. Purify the Cas9 mRNA and sgRNAs using the MEGAclear kit and elute with RNase-free water. C57BL/6 female mice and KM mouse strains were used as embryo donors and pseudopregnant foster mothers, respectively. Super-ovulated female C57BL/6 mice (3–4 weeks old) were mated to C57BL/6 stud males, and fertilized embryos were collected from the ampullae of super ovulated. Different concentrations of Cas9 mRNA and sgRNAs were mixed and co-injected into the cytoplasm of one-cell stage fertilized eggs. After injection, surviving zygotes were transferred into oviducts of KM albino pseudopregnant females. Sixty-five pups were born and then identified by genotyping with tail DNA. $Neat1^{+/-}$ mice were used furtherly to reproduce the next generation, which includes $Neat1^{-/-}$ mice and $Neat1^{+/+}$ mice. $IL$-$6^{-/-}$ mice were provided by professor Zexiong Lian's laboratory. Genotyping PCR primers sequences are shown in Supplementary Table 1. All animal experiments were undertaken and conducted with approval from the Animal Research Ethics Committee of the University of Science and Technology of China (Permit Number: USTCACUC1801057).

**Cell culture**. IBMDM, L929, and 293T cell lines were cultured in DMEM (Dulbecco's modified Eagle's medium) medium containing 10% fetal bovine serum. iBMDM and L929 cell lines were kindly provided by professor Rongbin Zhou. 293T cell lines (ATCC CRL-3216) were purchased from the American Type Culture Collection (ATCC, Manassas, VA, USA). All cells were tested for mycoplasma contamination by Cell Culture Contamination Detection Kit (ThermoFisher).

**Cell stimulation**. Macrophages were cultured in DMEM complemented with 10% FBS, 1 mM sodium pyruvate, and 2 mM L-glutamine overnight, and the medium was changed to opti-MEM in the following morning. Cells were primed with LPS (100 ng/mL) for 3 h, and subsequently stimulated with nigericin (5 μM) for 0.5 h to activate NLRP3 inflammasome, DOTAP-transfected flagellin (0.5 μg/mL) for 3 h to activate NLRC4 inflammasome, Lipofectamine 2000-transfected poly(dA:dT) (1 μg/mL) for 3 h to activate AIM2 inflammasome or 2DG (50 mM) for 4 h to activate Nlrp1b inflammasome. Total cell lysates, precipitated supernatants, and pellets of cells at different stages were analyzed by immunoblotting.

**CLIP sequencing**. CLIP was performed as previously described[67,68]. Briefly, two 10-cm dishes of iBMDM cells were stimulated with LPS for 3 h followed by treated with nigericin for 0.5 h. Activated iBMDMs were washed three times with cold PBS, and irradiated at 150 mJ/cm² at 254 nm in HL-2000 HybriLinker™ UV Crosslinker. Cells were collected and resuspended in 1 ml lysis buffer (50 mM Tris–HCl [pH 7.4], 100 mM NaCl, 1% Igepal CA-630, 0.1% SDS, 0.5% sodium deoxycholate, 1/25 volume of Roche Protease Inhibitor Cocktail, 40 U/mL RNase inhibitor, 20 μM MG132) for 20 min Cell lysates were precleared with protein A/G beads (Pierce), and then were incubated with protein A/G beads coated with NLRP3 antibody at 4 °C for 3 h. After extensive washing with washing buffer (50 mM Tris–HCl [pH 7.4], 100 mM NaCl, 0.2% Igepal CA-630, 0.1% SDS, 0.5% sodium deoxycholate, 40 U/mL RNase inhibitor), the bead-bound immunocomplexes were eluted using elution buffer (50 mM Tris-HCl, pH 8.0, 1% SDS, and 10 mM EDTA) at 65 °C for 10 min. To isolate protein-associated RNAs from the eluted immunocomplexes, samples were treated with proteinase K, and RNAs were extracted by phenol/chloroform. The RNA sequencing was performed and analyzed by KangChen Bio-tech, Shanghai, China. The sequencing data were deposited in the National Center for Biotechnology Information Gene Expression Omnibus database (GSE118722). Visualization analysis was performed by Integrative Genomics Viewer (IGV).

**Propidium iodide staining and imaging**. To examine pyroptotic cells, cells were treated as indicated in 12-well plates for image capture. Propidium iodide (1 μg/mL) was added to the medium for monitoring cell membrane integrity. Static bright-field images of pyroptotic cells were captured using an Olympus IX71.

**Immunoprecipitation and RNA pull-down**. For immunoprecipitation assay, the lysates of iBMDMs were incubated with 2 μg/ml antibody recognizing particular protein or normal rabbit/mouse IgG at 4 °C for 2 h, and then with 20 μl protein A/G magnetic beads for 1 h. After washing three times with lysis buffer (50 mM Tris–HCl [pH 7.4], 100 mM NaCl, 0.5% NP-40, 0.5% sodium deoxycholate, 1/25 volume of Roche Protease Inhibitor Cocktail, 40 U/ml RNase inhibitor, and 20 μM MG132), the immunocomplexes were analyzed by western blotting and real-time RT-PCR. For RNA pull-down with biotin-labeled DNA probes, cell lysates were incubated with biotinylated sense or antisense DNA oligomers (1 μM, Supplementary Table 1) corresponding to *Neat1* for 2 h, and then with 20-μl streptavidin-coupled agarose beads for 1 h. After extensive washing, the precipitated complexes were analyzed by real-time RT-PCR and western blotting. For RNA pull-down with in vitro biotin-labeled *Neat1_1*, 3-μg in vitro-synthesized biotin-labeled *Neat1_1* was incubated with lysates from indicated iBMDM cells (4 × 10⁷) for 3 h. Streptavidin-coupled agarose beads (Invitrogen) were then added to the reaction mix to isolate the RNA–protein complex. For RNA immunoprecipitation (RIP), indicated cell lysates were precleared with 20-μl protein A/G beads, and then incubated with 20-μl protein A/G beads coated with indicated antibodies at 4 °C for 2 h. After extensive washing and elution, immunocomplexes were analyzed by real-time RT-PCR and western blotting. Primers used for template amplification are shown in Supplementary Table 1.

**RNA fluorescence in situ hybridization following by immunofluorescence**. In order to detect *Neat1* and ASC in a same cell, RNA FISH was firstly carried out as previously described[69] with in vitro transcribed antisense probes (Supplementary Table 1) labeled by Nucleic Acid Labeling Kits (Life technologies, USA) with Alexa Fluor 488. Then fluorescence probes-labeled cells were stained with rabbit anti-ASC and Alexa fluor 647 donkey anti-rabbit IgG as described in the next item.

**Immunofluorescence**. IBMDMs, seeded in glass-bottom dishes were treated with indicated conditions. The cells were fixed with 4% paraformaldehyde in PBS for 30 min, permeabilized with 0.5% Triton X-100 in PBS for 15 min, rinsed, blocked with 5% BSA in PBS for 30 min, and then incubated with indicated primary

antibodies overnight at 4 °C. the next morning, cells were washed three times with PBST (0.05% Tween-20 in PBS) and were recognized with indicated fluorescence-labeled secondary antibodies. After extensive washing with PBST, cells were imaging.

**Pellets preparation and western blotting**. Cells were harvested after LPS and nigericin co-treatment, resuspended in 0.3 mL buffer (20 mM Hepes [pH 7.5], 150 mM KCl, 1.5 mM MgCl₂, 1 mM EGTA, 1 mM EDTA, 320 mM sucrose, 1/25 volume of Roche Protease Inhibitor Cocktail, 40 U/mL RNase inhibitor, 20 μM MG132) lysed by shearing 10 times through a 27-gauge syringe. After centrifugation (520 × g, 8 min), supernatants were mixed with equal volumes of CHAPS buffer (20 mM Hepes-KOH [pH 7.5], 5 mM MgCl₂, 0.5 mM EGTA, and 0.1% CHAPS) and centrifuged (4000 × g) for another 8 min to pellet the ASC pyroptosomes. The pellets then were resuspended in CHAPS buffer containing 2 mM disuccinimidyl suberate (DSS) and incubated for 30 min followed by centrifugation (4000 × g, 10 min). Pellets were fractionated on 12% SDS/PAGE and detected by immunoblotting with ASC antibody. Furthermore, western blotting was performed by the method described previously[70].

**Enzyme-linked immunosorbent assay**. Supernatants from cell culture were assayed for mouse IL-1β and mouse TNF-α according to manufacturer's instructions of ELISA kits (R&D).

**Cytosolic and nuclear fractionation**. Indicated cells were incubated with hypotonic buffer (25 mM Tris-HCl, pH 7.4, 1 mM MgCl₂, 5 mM KCl) on ice for 5 min. An equal volume of hypotonic buffer containing 1% NP-40 was then added, and each sample was left on ice for another 5 min. After centrifugation at 5000 × g for 5 min, the supernatant was collected as the cytosolic fraction. The pellets were resuspended in nucleus resuspension buffer (20 mM HEPES, pH 7.9, 400 mM NaCl, 1 mM EDTA, 1 mM EGTA, 1 mM DTT, 1 mM PMSF) and incubated at 4 °C for 30 min. Nuclear fraction was collected after removing insoluble membrane debris by centrifugation at 12,000 × g for 10 min.

**Real-time RT-PCR**. Total RNA was isolated by TRIzol reagent (Invitrogen). One microgram of RNA was used to synthesize cDNA using the RT Master Mix (Takara). Real-time PCR was performed using SYBR Green real-time PCR analysis (Takara) with the specific primers (Supplementary Table 1). PCR results, recorded as cycle threshold (Ct), were normalized against an internal control (*Actin*).

**In vivo peritonitis**. Wild-type or *Neat1* knockout mice (males, 8 weeks old) were i. p. injected with 35 mg/kg Alum for 12 h. Peritoneal cavities were washed with 6 mL of PBS. The peritoneal fluids were harvested and concentrated for ELISA analysis with Amicon Ultra 10 K from Millipore. The concentrations of mouse IL-1β were measured using ELISA kits (R&D) according to the manufacturer's instruction. PECs were analyzed by flow cytometry[56].

**In vivo pneumonia**. *Neat1*⁺/⁺ or *Neat1*⁻/⁻ mice (males, 8 weeks old) were intranasally instilled with a mixture of flagellin (1 μg per mouse) and TurboFect in vivo transfection reagent in a total volume of 50 μL for 12 h to induce pneumonia or with isotonic saline for 12 h as negative control. BALF was obtained by lavaging the lung with 3 mL PBS containing soybean trypsin inhibitor (100 μg/mL). The concentrations of mouse IL-1β were measured using ELISA kits (R&D). BALCs in BALF were analyzed by flow cytometry.

**Flow cytometry**. PECs and BALCs were obtained and blocked in PBS with 20% rat serum for 0.5 h. After washing three times with PBS, cells were incubated with PerCP-CyTM5.5-mouse CD11b antibody, APC-mouse Ly-6G antibody, and PE-mouse Ly-6C antibody for 1 h. After washing three times with PBS, stained cells were analyzed on the BD FACSCalibur to detect the proportion of neutrophils or Ly-6C⁺ monocytes. The flow cytometry data were analyzed on FlowJo software with the gating strategies indicated in Supplementary Fig. 10.

**shRNAs and siRNAs**. Forward (F) and reverse (R) DNA fragments (Supplementary Table 1) were annealed and then cloned into plasmid pLKO.1, which was used for gene knockdown in iBMDMs. siRNAs were used for gene knockdown in BMDMs. siRNAs' target sequences are shown in Supplementary Table 1.

**Reporting Summary**. Further information on experimental design is available in the Nature Research Reporting Summary linked to this article.

## Data availability

All relevant data are available from the corresponding authors upon reasonable request. The source data underlying all reported averages in Figs. 1b, c, e, g, i, 4a–d, 5b, d, e, f, 6b, d, f, 7a–c, 8g, h, 10a–h and Supplementary Figs. 1c–i, k, l, 3b–e, 4a–h, 7a–g, 8f, 9a are provided in Source Data File. The source data underlying all uncropped versions of the blots and gels presented in Figs. 1a, d, f, h, 2a–h, 3a–d, 4a–f, 5a, c, 8a–c, e, f, 9a–i and

Supplementary Figs. 1j, m, 2a, b, 3a, 4f, h, 5d, 6a–e, 8a, d, e, 9b are also provided in Source Data File. The sequencing data were deposited in the National Center for Biotechnology Information Gene Expression Omnibus database (GSE118722).

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

## Acknowledgements

We would like to thank Professor Zexiong Lian (South China University of Technology) for kindly providing $IL-6^{-/-}$ mice. This work was supported by grants from the National Key R&D Program of China (2018YFA0107103 and 2016YFC1302302) and the National Natural Science Foundation of China (81430065 and 81820108021) to M.W. and the U.S. National Institutes of Health (R01CA182675 and R01CA184867) to X.Y.

## Author contributions

P.Z., L.C., X.Y. and M.W. designed research; P.Z. and L.C. performed experiments and analyzed data; R.Z. helped to initiate this project and provided material support; P.Z., X.Y. and M.W. wrote and revised the manuscript.

## Additional information

**Competing interests:** The authors declare no competing interests.

