## [Peer Review File · Nature Communications]

Reviewers' comments:

Reviewer #1 (Inflammasome signaling and function)(Remarks to the Author):

This manuscript by Zhang, Wu and colleagues reports the discovery of lncRNA Neat1 as an essential regulator of canonical inflammasome (NLRP3, NLRC4, AIM2) assembly through stabilization of the mature caspase-1 tetramers. Interestingly, NLRP3 inflammasome stimulation leads to disruption of paraspeckle, which in turn facilitates the translocation of Neat1 from the nucleus to the cytoplasm to mediate inflammasome assembly. This may represent a regulatory mechanism for the majority of inflammasomes involving caspase-1. In addition, both human and murine Neat1 mediate inflammatory response to hypoxia through HIF2a-induced expression of Neat1. Lastly, the authors demonstrate the in vivo function of Neat1 as an NLRP3 inflammasome regulator through a murine model of peritonitis. These are novel findings that will be of interest to the inflammasome field.

Major issues:

The main issue resides with whether Neat1 translocation is a general mechanism for inflammasome activation. The translocation of Neat1 upon LPS and nigericin treatment and paraspeckle disruption is potentially an important finding. Since the authors demonstrated that Neat1 is also important for the activation of the NLRC4 and AIM2 inflammasomes, do we know if flagellin and poly(dA:dT) treatment of cells also disrupt paraspeckle and induce Neat1 translocation? What about other caspase-1 dependent inflammasomes such as NLRP1?

The role of Neat1 as a mediator of the NLRP3 inflammasome activation upon hypoxia stress is a very interesting finding. However, this provides no mechanistic insights for the regulation of other inflammasomes (NLRC4 and AIM2) regulated by Neat1, but may not be responsive to hypoxia stress.

The mechanisms of Neat1 in regulating inflammasome activation is not clear from the data presented here. Figure 4G is referred to as evidence that Neat1 increased the interaction between p20 and p10. However, difference in the intensity of the bands marked with an arrow in "IB: HA antibody" panel is very minor. It is hardly convincing that such minor difference in p20:p10 association attributed to Neat1 could contribute to the dramatic effects of Neat1 on IL-1b release and pyroptosis in figures 1-2 and 5.

Minor issues:

Some of the arrows have no labels. For example figure 4G and figure S5F.

Reviewer #2 (lncRNA and signalling)(Remarks to the Author):

The study entitled "The lncRNA Neat1 promotes activation of inflammasomes in macrophages" investigates the role of the lncRNA Neat1 in modulating the "inflammasome" response. The authors find that, in mouse bone-marrow derived and immortalized macrophages, Neat1 not only facilitates the complex formation between NLRP3, NLRC4, and AIM2 and stabilization of mature caspase-1 and in turn interleukin-1 β release followed by cell death. Interestingly, Neat1 is translocated into the cytoplasm upon stimulation of the inflammasome; something that, to this reviewer's knowledge, is the first time this has been reported in such mechanistic detail. Finally, the authors take their findings to test them in vivo and find that Neat1 loss of function reduces the inflammation response. Collectively,

these results demonstrate a convincing role for Neat1 in mediating specified inflammasome responses.

1) Interestingly, the inflammasome is assembled in the cytoplasm thus requiring shuttling of Neat1 to the cytoplasm, something not frequently reported. The authors show that Neat1 RNA goes from 1% to 8% cytoplasmic upon stimulation. However, the RNA-FISH images do not resolve this same 8 fold shift at a sufficiently convincing level. The images in figure 6C do not show a compelling "cytoplasmic" localization. It is strongly suggested that these RNA-FISH analyses be done at a more compelling resolution and or with co-localization of ASC components.

This is very important as the mechanism is taking place in a region that Neat1 does not (or to this reviewers knowledge has ever been observed) typically localize. Thus, if this mechanism is taking place in the cytoplasm as all the evidence points to in this study, then it requires due diligence in strong supporting RNA-FISH data to see colocalization of Neat1 and inflammasomes in the cytoplasm. Yet at the very least much more compelling RNA-FISH data and images to demonstrate this unusual property of Neat1.

In short, considering how much depends on the cytoplasmic localization of Neat1 it should be much more rigorously determined by RNA-FISH.

2) The authors set up the context that the inflammasome has to respond in many ways how does neat1 specify a new aspect or specialization of this pathway. For example, inflammasome response results in "ASC" formation and I believe implied that the ASC recruits additional factors for specific responses. They identify Neat1 by RNA immunoprecipitation of NLRP3 (as well as Gm26917, Gm42418 {appear to be pseudogenes located on Christmas 17}, Gm26767 a poorly aligned region of the genome) followed by RNA-sequencing.

It would seem reasonable to see if NLRC4 and AIM2 RNA co-immunoprecipitates with Neat1 as well? Thus still leaving a mystery of how these inflammasomes are assembled independent of Neat1 in the NLRP3 inflammasome?

Another important relative context: in the Neat1^{-/-} mouse models how are the NLRC4 and AIM2 responses altered? The authors have an interesting set up as to how Neat1 may mediate specific responses, but with the majority of focus on NLRP3 inflammasome it is difficult to know if it is unique to this ASC or works in AIM2 and NLRC4 inflammasome responses as well? The authors show that Neat1 also activates these complexes, but it isn't clear how Neat1 is specifying "unique" responses relative to the NLRP3 inflammasome.

Does Neat1 facilitate Caspase-1 activity in the NLRC4 and AIM2 response as well? If so maybe it is not a "specification" factor but rather universal inflammasome factor that facilitates a multi-step recruitment, maturation and stabilization of Caspase-1 in all ASCs?

Response to reviewers:

We thank reviewers for both their enthusiasm for our work and the thoughtful and constructive comments to further improve it. Accordingly, we have performed a number of additional experiments and extensively revised the manuscript, which we believe results in a much-improved manuscript. Our point-by-point response to reviewers' critiques are provided below. Please note the newly added and modified parts of text are highlighted in red fonts for easier reference.

Response to Reviewer #1:

This manuscript by Zhang, Wu and colleagues reports the discovery of lncRNA Neat1 as an essential regulator of canonical inflammasome (NLRP3, NLRC4, AIM2) assembly through stabilization of the mature caspase-1 tetramers. Interestingly, NLRP3 inflammasome stimulation leads to disruption of paraspeckle, which in turn facilitates the translocation of Neat1 from the nucleus to the cytoplasm to mediate inflammasome assembly. This may represent a regulatory mechanism for the majority of inflammasomes involving caspase-1. In addition, both human and murine Neat1 mediate inflammatory response to hypoxia through HIF2a-induced expression of Neat1. Lastly, the authors demonstrate the in vivo function of Neat1 as an NLRP3 inflammasome regulator through a murine model of peritonitis. These are novel findings that will be of interest to the inflammasome field.

Major issues:

The main issue resides with whether Neat1 translocation is a general mechanism for inflammasome activation. The translocation of Neat1 upon LPS and nigericin treatment and paraspeckle disruption is potentially an important finding. Since the authors demonstrated that Neat1 is also important for the activation of the NLRC4 and AIM2 inflammasomes, do we know if flagellin and poly(dA:dT) treatment of cells also disrupt paraspeckle and induce Neat1 translocation? What about other caspase-1 dependent inflammasomes such as NLRP1?

Response: As suggested by the reviewer, we have now performed several experiments and found that more *Neat1* was released from the nucleus to the cytoplasm when the NLRC4, AIM2 and Nlrp1b inflammasomes were activated in iBMDMs by flagellin, poly(dA:dT) and 2DG, respectively (Fig. 7b, c and Supplementary Fig. 7d-g). In macrophages, flagellin, poly(dA:dT) and 2DG treatment also decreased levels of paraspeckle proteins (Fig. 8b, c and Supplementary Fig. 8a). Moreover, we have analyzed the interactions among paraspeckle components in untreated, LPS-priming, and flagellin-, poly(dA:dT)- or 2DG-activated iBMDMs. We observed that interaction of Nono with Pspc1 or Sfpq was significantly inhibited in response to stimuli that activate the NLRC4, AIM2 and Nlrp1b inflammasomes (Fig. 8e, f and Supplementary Fig. 8d). These results indicate that activation of NLRC4, AIM2 and Nlrp1b inflammasomes is also associated with the disruptions of paraspeckles and the *Neat1* translocation from these nuclear bodies to the cytoplasm. Some sentences have been added in text to indicate these results (page 11 and 12).

The role of Neat1 as a mediator of the NLRP3 inflammasome activation upon hypoxia stress is a very interesting finding. However, this provides no mechanistic insights for the regulation of other inflammasomes (NLRC4 and AIM2) regulated by Neat1, but may not be responsive to hypoxia stress.

Response: To investigate whether the hypoxic stress affects other inflammasomes, we compared the activation of NLRC4 and AIM2 inflammasomes in iBMDMs under hypoxic and normoxic conditions. Interestingly, the assembly of the NLRC4 or AIM2 inflammasomes, as well as caspase-1 activation and IL-1 β secretion, were increased when iBMDMs were cultured under the hypoxic conditions (Fig. 9b and c). This is dependent on HIF-1 α and HIF-2 α , as knocking down either one of them inhibited caspase-1 activation and IL-1 β maturation under the hypoxic conditions (Fig. 9f-i).

According to these results, we assumed that hypoxic stress might regulate NLRC4 and AIM2 inflammasome in a HIF-2 α -induced *Neat1*-mediated manner. To test this hypothesis, we silenced HIF-2 α and *Neat1* individually or in combination.

Indeed, in iBMDMs devoid of *Neat1*, knockdown HIF-2 α did not further reduce the activation of diverse inflammasomes, while knocking down HIF-1 α did (Fig. 9f-i). These data indicate that, like the NLRP3 inflammasome, hypoxia stress activates the NLRC4 and AIM2 inflammasomes via HIF-2 α -mediated up-regulation of *Neat1*. These data are now a part of the result in this revision (page 13).

The mechanisms of Neat1 in regulating inflammasome activation is not clear from the data presented here. Figure 4G is referred to as evidence that Neat1 increased the interaction between p20 and p10. However, difference in the intensity of the bands marked with an arrow in “IB: HA antibody” panel is very minor. It is hardly convincing that such minor difference in p20:p10 association attributed to Neat1 could contribute to the dramatic effects of Neat1 on IL-1 β release and pyroptosis in figures 1-2 and 5.

Response: *Neat1* promotes caspase-1 activation likely through at least two mechanisms. First, it increases the interaction of pattern recognition receptors (PRRs)—such as NLRC4, AIM2 and NLRP3—with ASC and the recruitment of pro-caspase-1 (Fig. 2g, h and Supplementary Fig. 1m). For example, *Neat1* increased the interaction of NLRP3 with ASC by ~70% and the recruitment of pro-caspase-1 to the NLRP3 inflammasomes by ~80% (Supplementary Fig. 1m). This led to substantial enhancement in the processing of pro-caspase-1 to generate the mature, tetrameric form (Fig. 1f). Second, *Neat1* also increase inter-subunit interactions of the mature caspase-1, enhancing its protease activity. For example, *Neat1* increase the association of p10 and p20 by ~100%, and the association of p10 with p33 by ~110%, in co-immunoprecipitation assays (Fig. 5a, c and Supplementary Fig. 6b-e). These values are obtained from three independent experiments. The band intensity was quantified by ImageJ and displayed numerically in histograms (Fig. 5b, d). Collectively, the augmented processing of pro-caspase-1 and the stabilization of the mature caspase-1 tetramer likely result in a strong increase in IL-1 β release and

pyroptosis (Figure 1G and 6B). These data are now a part of the result in this revision (page 10).

Minor issues:

Some of the arrows have no labels. For example figure 4G and figure S5F.

Response: We have now added labels for these and other arrows including those in Fig. 5a, c and Supplementary Fig. 5d, 6b-e.

Response to Reviewer #2:

The study entitled "The lncRNA Neat1 promotes activation of inflammasomes in macrophages" investigates the role of the lncRNA Neat1 in modulating the "inflammasome" response. The authors find that, in mouse bone-marrow derived and immortalized macrophages, Neat1 not only facilitates the complex formation between NLRP3, NLRC4, and AIM2 and stabilization of mature caspase-1 and in turn interleukin-1 β release followed by cell death. Interestingly, Neat1 is translocated into the cytoplasm upon stimulation of the inflammasome; something that, to this reviewer's knowledge, is the first time this has been reported in such mechanistic detail. Finally, the authors take their findings to test them in vivo and find that Neat1 loss of function reduces the inflammation response. Collectively, these results demonstrate a convincing role for Neat1 in mediating specified inflammasome responses.

1) Interestingly, the inflammasome is assembled in the cytoplasm thus requiring shuttling of Neat1 to the cytoplasm, something not frequently reported. The authors show that Neat1 RNA goes from 1% to 8% cytoplasmic upon stimulation. However, the RNA-FISH images do not resolve this same 8 fold shift at a sufficiently convincing level. The images in figure 6C do not show a compelling "cytoplasmic" localization. It is strongly suggested that these RNA-FISH analyses be done at a more compelling resolution and or with co-localization of ASC components.

This is very important as the mechanism is taking place in a region that *Neat1* does not (or to this reviewer's knowledge has ever been observed) typically localize. Thus, if this mechanism is taking place in the cytoplasm as all the evidence points to in this study, then it requires due diligence in strong supporting RNA-FISH data to see colocalization of *Neat1* and inflammasomes in the cytoplasm. Yet at the very least much more compelling RNA-FISH data and images to demonstrate this unusual property of *Neat1*.

In short, considering how much depends on the cytoplasmic localization of *Neat1* it should be much more rigorously determined by RNA-FISH.

Response: We thank the reviewers for these comments and suggestions with regard to *Neat1*-FISH and ASC-IF. Previously, we performed *Neat1*-RNA-FISH (excited with 488 nm laser) together with nuclear DNA staining by propidium iodide (excited with 532 nm laser). However, these two signals interfere with each other. To perform RNA-FISH at a more compelling resolution, we have now performed RNA-FISH combined with IF to detect not only the cytoplasmic localization of *Neat1* but also the co-localization of *Neat1* and ASC in untreated, LPS-primed, and LPS-nigericin-activated iBMDMs. As shown in Fig. 7d, cytoplasmic localization of *Neat1* and the co-localization of *Neat1* and ASC specks were only observed in LPS-nigericin-activated iBMDMs. This observation is consistent with the notion that, in the activated iBMDMs, *Neat1* is translocated from nucleus into cytoplasm where it interacts with, and promotes the formation of, the NLRP3 inflammasome. These results have been included in the revised manuscript (page 11).

2) The authors set up the context that the inflammasome has to respond in many ways how does *neat1* specify a new aspect or specialization of this pathway. For example, inflammasome response results in "ASC" formation and I believe implied that the ASC recruits additional factors for specific responses. They identify *Neat1* by RNA immunoprecipitation of NLRP3 (as well as *Gm26917*, *Gm42418* {appear to be

pseudogenes located on Christmas 17}, Gm26767 a poorly aligned region of the genome) followed by RNA-sequencing.

It would seem reasonable to see if NLRC4 and AIM2 RNA co-immunoprecipitates with Neat1 as well? Thus still leaving a mystery of how these inflammasomes are assembled independent of Neat1 in the NLRP3 inflammasome?

Response: Based on reviewer's comments, we have examined the association of *Neat1* with the NLRC4 and AIM2 inflammasomes. We performed *Neat1* pulldown in lysates of untreated iBMDMs, LPS-primed iBMDMs, and LPS/flagellin- or LPS/poly(dA:dT)-activated iBMDMs. As displayed in Supplementary Fig. 4e-h, *Neat1* also bound to pro-caspase-1 under all conditions, and with NLRC4, AIM2 and ASC only upon the activation of the NLRC4 and AIM2 inflammasomes. Furthermore, co-immunoprecipitation assays with ASC antibody shown that *Neat1* significantly enhanced the assembly of NLRC4 inflammasome and AIM2 inflammasome (Fig. 2g, h). These data indicate that in addition to the NLRP3 inflammasome, *Neat1* also interacts with both the NLRC4 and AIM2 inflammasomes.

Previous studies have shown that the expression of *Neat1* is up-regulated by various signals that also activates the inflammasomes. We have also found that cytoplasmic translocation of *Neat1* is induced by stimuli that activate the NLRP3, NLRC4, and AIM2 inflammasomes (Fig. 7a-c). Therefore, the up-regulation and translocation of *Neat1* may be a converging event leading to the activation of the various caspase-1-associated canonical inflammasomes by a wide range of stimuli. All these have been included in the results part (page 7, 8 and 11).

Another important relative context: in the *Neat1*^{-/-} mouse models how are the NLRC4 and AIM2 responses altered? The authors have an interesting set up as to how *Neat1* may mediate specific responses, but with the majority of focus on NLRP3 inflammasome it is difficult to know if it is unique to this ASC or works in AIM2 and NLRC4 inflammasome responses as well? The authors show that *Neat1* also activates

these complexes, but it isn't clear how *Neat1* is specifying "unique" responses relative to the NLRP3 inflammasome.

Response: Based on reviewer's comment, we have now used a mouse model of flagellin-induced pneumonia to evaluate the function of *Neat1* on NLRC4 inflammasome activation *in vivo*. As seen in Fig. 10e-h, *Neat1* deficient led to reduced IL-1 β secretion in bronchoalveolar lavage fluid (BALF) and decreased recruitment of inflammatory cells. So far as we know, there is no common mouse model to study AIM2 inflammasome activation *in vivo*. However, using BMDMs isolated from mice, we found that the loss of *Neat1* inhibited mature caspase-1 activity (Fig. 5e) and decreased IL-1 β secretion (Fig. 2f), suggesting that *Neat1* is likely to enhance the activation of AIM2 inflammasome *in vivo*. These new results extend out previous findings, and suggest that *Neat1* may be a common enabling factor for the assembly of several canonical inflammasomes in response to a diverse range of stimuli. We have included these in the result part of this revised manuscript (page 10 and 14).

Does *Neat1* facilitate Caspase-1 activity in the NLRC4 and AIM2 response as well? If so maybe it is not a "specification" factor but rather universal inflammasome factor that facilitates a multi-step recruitment, maturation and stabilization of Caspase-1 in all ASCs?

Response: Indeed, as shown in Fig. 5e and 5f, *Neat1* increased the protease activity of caspase-1 upon the activation of the NLRC4 and AIM2 inflammasomes. In addition, *Neat1* also enhanced the assembly of NLRC4 and AIM2 inflammasomes, the recruitment of pro-caspase-1, and the interaction of pro-caspase-1 with ASC (Fig. 2g, h). Due to the function of *Neat1* on inflammasome assembly, *Neat1* furtherly increased caspase-1 self-cleavage to generate more subunits for mature caspase-1 when diverse inflammasomes were activated (Fig. 1a, d, f, h, 2a-f and Supplementary Fig. 2a, b). Moreover, *Neat1* strengthened subunit-subunit interactions of caspase-1

tetramers (Fig. 5a-d), and increased the protease activity of caspase-1 (Fig. 5e, f). Again, these data suggest that *Neat1* is a general factor for various inflammasomes in activated macrophages, acting at multiple steps during the recruitment and activation of caspase-1. This information has been added in this revision (page 7, 10 and 15).

REVIEWERS' COMMENTS:

Reviewer #1 (Remarks to the Author):

The revised manuscript has sufficiently addressed the reviewer's concerns, and the manuscript should be suitable for publication.

Reviewer #2 (Remarks to the Author):

I thank the authors for addressing my major concerns and many of the others raised by the reviewers. Specifically the RNA FISH and RNA-protein interaction data greatly strengthen this interesting study. I have no further suggestions for the authors.

Response to Reviewer #1:

The revised manuscript has sufficiently addressed the reviewer's concerns, and the manuscript should be suitable for publication.

Response: Many thanks for reviewer #1's comment.

Response to Reviewer #2:

I thank the authors for addressing my major concerns and many of the others raised by the reviewers. Specifically the RNA FISH and RNA-protein interaction data greatly strengthen this interesting study. I have no further suggestions for the authors.

Response: We appreciate reviewer #2's positive comment.